



Atmospheric
Chemistry
and Physics

# Secondary organic aerosol formation from photooxidation of furan: effects of NO$_x$ and humidity

**Xiaotong Jiang**[1], **Narcisse T. Tsona**[1], **Long Jia**[2], **Shijie Liu**[1], **Hailiang Zhang**[2], **Yongfu Xu**[2], and **Lin Du**[1]

[1]Environment Research Institute, Shandong University, Qingdao, 266237, China
[2]State Key Laboratory of Atmospheric Boundary Layer Physics and Atmospheric Chemistry,
Institute of Atmospheric Physics, Chinese Academy of Sciences, Beijing, 100029, China

**Correspondence:** Lin Du (lindu@sdu.edu.cn)

**Abstract.** Atmospheric furan is a primary and secondary pollutant in the atmosphere, and its emission contributes to the formation of ultrafine particles. We investigate the effects of NO$_x$ level and humidity on the formation of secondary organic aerosol (SOA) generated from the photooxidation of furan in the presence of NaCl seed particles. SOA mass concentration and yield were determined under different NO$_x$ and humidity levels. A significant difference is observed both in the variation of SOA mass concentration and SOA yield with the initial experimental conditions. Varying VOC (volatile organic compound) / NO$_x$ ratios over the range 48.1 to 8.2 contributes to the effective formation of SOA in the presence of NaCl seed particles, with the SOA mass concentration and SOA yield ranging from 0.96 to 23.46 µg m$^{-3}$ and from 0.04 % to 1.01 %, respectively. We found that there was a favourable relationship between the SOA yields and NO$_x$ concentration. In particular, the increase in SOA yield with increasing NO$_x$ concentration was continuously observed at high NO$_x$ levels owing to a corresponding increase in the amount of low-volatility hydroxyl nitrates and dihydroxyl dinitrates that can partition into the particle phase. In addition, varying relative humidity (RH) from 5 % to 88 % increased the SOA yield from 1.01 % to 5.03 %. The enhanced SOA formation from humid conditions may result from the high OH concentration, rapid furan decay rate, enhanced carbonyl-rich products condensation, and the aqueous-phase reactions. Using hybrid quadrupole-orbitrap mass spectrometer equipped with electrospray ionization (HESI-Q Exactive-Orbitrap MS), three carbonyl-rich products and three kinds of organonitrates were identified in the collected SOA. Based on the HESI-Q Exactive-Orbitrap MS analysis and Fourier transform infrared spectroscopy (FTIR), the reaction mechanism of furan photooxidation was proposed. This study demonstrates the effects of NO$_x$ and humidity on SOA formation during the furan–NO$_x$–NaCl photooxidation and provides new insights into the oxidation regime and SOA composition in furan photooxidation. The results also illustrate the importance of studying SOA formation over a comprehensive range of environmental conditions. Only such evaluations can induce meaningful SOA mechanisms to be implemented in air quality models.

## 1 Introduction

Atmospheric particulate matter (PM) is primarily composed of organic carbon, elemental carbon, sulfate, nitrate, and other components (Donahue et al., 2009; Zhang et al., 2011), which have adverse effects on human health and global climate forcing (Hallquist et al., 2009; Pope III et al., 2013). Secondary organic aerosols (SOAs) constitute a substantial portion of the total ambient aerosol particles, which mainly originate from biomass burning and atmospheric reactions of volatile organic compounds (VOCs) (Kanakidou et al., 2005). Identifying the chemical composition and major precursors of SOA is helpful to better understand their formation mechanism and strategies for the control of PM. Furan is an important five-member heteroatom-containing VOC present in the atmosphere, and it can be produced by wood combustion, oil refining, coal mining and gasification, and pyrolysis of biomass, cellulose, and lignin (Shafizadeh, 1982). As shown in previous studies, furan is one of the most com-

mon products of the thermal cracking of biomass (Kahan et al., 2013; Gilman et al., 2015). Both furan and furan-related compounds have been detected in the effluent from initial smouldering of fuels, oats, and soft pellets (Olsson, 2006; Perzon, 2010). In addition, furan has been identified as one of the major compounds emitted from the combustion of ponderosa pine, which constitute a significant fraction (5 %–37 % by estimated emission factor) of smoke from combustion (Hatch et al., 2015). Field measurements of hydrocarbon emissions from biomass burning in Brazil have also shown that furans consist of 52 % and 72 % of the oxygenated hydrocarbons emissions in the *Cerrado* (grasslands) and selva (tropical forest) regions, respectively (Greenberg et al., 1984). Moreover, furan has been proven to be the typical trace gas of roasting or burning activities (Gloess et al., 2014; Coggon et al., 2016; Burling et al., 2010; Gilman et al., 2015; Sarkar et al., 2016; Stockwell et al., 2016, 2015, 2014), and it is also an important contribution to OH reactivity towards biomass burning emissions (Gilman et al., 2015; Sarkar et al., 2016). Furan is also a secondary pollutant produced in the photooxidation of some conjugate alkenes. Previous product studies of 1,3-butadiene oxidation have shown that furan is one of the dominant product during the OH radical-initiated reactions in the presence of NO (Sprengnether et al., 2002; Tuazon et al., 1999). Furthermore, another source of furan is the cyclization of the unsaturated 1,4-hydroxyaldehydes, which will further go through the dehydration reactions (Baker et al., 2005). There is strong evidence that furan can cause malignant tumour formation in a variety of experimental animals (IARC, 1995). Therefore, a better understanding of the atmospheric transformations of furan will be helpful to improve the air quality models for toxic species.

As an aromatic hydrocarbon, furan is very reactive according to its electrophilic substitution reactions (Villanueva et al., 2007). The measured rate constant of furan with OH radicals, $O_3$, and $NO_3$ radicals at room temperature is $(4.01 \pm 0.30) \times 10^{-11}$, $(2.42 \pm 0.28) \times 10^{-18}$, and $(1.4 \pm 0.2) \times 10^{-12}$ $cm^3$ molecule$^{-1}$ s$^{-1}$, respectively (Atkinson et al., 1983, 1985). If the oxidant concentration of OH and $NO_3$ is assumed to be $1.6 \times 10^6$ and $5 \times 10^8$ molecules cm$^{-3}$, lifetimes of furan for reactions with OH and $NO_3$ are estimated to be 3 h and 18 min, respectively (Cabanas et al., 2004). In the troposphere, atmospheric furan is expected to be mainly oxidized by OH during the daytime and $NO_3$ at night. ~~Although the determination of kinetics and products of furan oxidation has been performed (Cabanas et al., 2004; Liljegren and Stevens, 2013; Tapia et al., 2011), the influence of several factors including $NO_x$ level and relative humidity on SOA formation from furan has not been well examined.~~ CEI It is important to assess the SOA formation potential of furan and its role in SOA production in biomass burning plumes. Although the SOA formation potential of furan has been studied in recent years (Cabanas et al., 2004; Liljegren and Stevens, 2013; Tapia et al., 2011), the influence

of several factors including $NO_x$ levels and relative humidity (RH) on SOA formation should be further investigated. It is generally accepted that $NO_x$ level plays a critical role in SOA formation, by governing the reactions of organo-peroxy radicals ($RO_2$) (Song et al., 2007, 2005). The branching ratio of $RO_2$ reactions with $NO_x$ and the hydroperoxyl radical ($HO_2$) is determined by the $NO_x$ level during experiments (Kroll and Seinfeld, 2008). A previous study showed that the mechanisms of the two reactions differ sharply under different $NO_x$ level conditions. $RO_2$ reacts only with NO under high $NO_x$ levels and with $HO_2$ under low-$NO_x$ conditions (Ng et al., 2007a). The latter case produces lower-volatility products than the former one. An increase in SOA yield with increasing $NO_x$ was proposed to be due to the formation of low-volatility compounds including multifunctional nitrates and dinitrates, which partitioned to the particle phase and contribute significantly to isoprene SOA under high-$NO_x$ conditions (Schwantes et al., 2019). Another study of the SOA formation from the irradiation of propylene also showed that the SOA yield decreased with increasing propylene / $NO_x$ ratio (Ge et al., 2017a). Recently, C. Liu et al. (2019) indicated that $NO_2$ could participate in the OH-initiated reaction of guaiacol, consequently resulting in the formation of organic nitrates and promoting SOA formation (C. Liu et al., 2019). The effect of $NO_x$ level on SOA formation from aromatic precursors has been investigated previously, but the results are inconclusive. A study focusing on the photooxidation of toluene and *m*-xylene has demonstrated that aerosol yields decrease as $NO_x$ level increases (Xu et al., 2015), while another study showed that the SOA yield from photooxidation of isoprene under high $NO_2/NO$ is 3 times more important than that measured under low $NO_2/NO$ (Chan et al., 2010).

Besides the mixed effect of $NO_x$ level on SOA formation, the RH can significantly alter the sizes of SOA particles, depending on their hygroscopicity (Varutbangkul et al., 2006), their gas-phase reactions involving water (Jonsson et al., 2006), and the aqueous chemistry occurring at their surfaces (Lim et al., 2010; Grgic et al., 2010). The photooxidation experiments with aromatic compounds, such as toluene (Edney et al., 2000; Cao and Jang, 2010; Faust et al., 2017; Hinks et al., 2018), benzene and ethylbenzene (Jia and Xu, 2014), *m*-xylene (Zhou et al., 2011), and 1,3,5-trimethylbenzene (Cocker et al., 2001) have been carried out previously to study the effect of RH on SOA formation. The results exhibited large discrepancies under different experimental conditions. The yield of SOA generated under low-$NO_x$ conditions was found to be small at high RH (Cao and Jang, 2010). A little RH effect was also observed on SOA particle formation and size distributions (Bonn and Moortgat, 2002; Fry et al., 2009). Nevertheless, positive correlations between RH and SOA yield have also been observed in the presence of hygroscopic seed particles (Zhou et al., 2011), and the role of RH in SOA formation also appears to be mixed. Water vapour not only participates in the VOC photooxidation reactions and affects the gas-phase oxidation mechanisms but also al-

ters the partitioning of the reaction products between the gas phase and the particle phase, thereby influencing the equilibrium partitioning of generated organic aerosols (Spittler et al., 2006). Moreover, the RH is connected with SOA formation due to its strong influence on seed acidity and particle liquid water concentration (Mahowald et al., 2011). A high-RH environment contributes to the increase in aerosol liquid water (ALW), which promotes the hydrolysis of organic compounds and leads to other heterogeneous and aqueous-phase reactions (Ervens et al., 2011). Previous studies regarding the atmospheric reactions of furan typically focused on the kinetics and mechanism (Gomez Alvarez et al., 2009; Aschmann et al., 2014).

In the present study, we used FTIR, in concert with a scanning mobility particle sizer (SMPS) to elucidate the roles of $NO_x$ level and RH in SOA formation from furan–$NO_x$ irradiation. All the experiments were conducted in the presence of NaCl seed particles, which acted as the nuclei and provided sufficient seed surface area at the beginning of the reaction to suppress the effects of vapour wall losses of semi- or low-volatility species. Specifically, we evaluate whether the increased ALW affects the SOA mass concentration and the SOA yield. The chemical composition of furan SOA was investigated by heated electrospray ionization high-resolution orbitrap mass spectrometer (HESI-Q Exactive-Orbitrap MS), with a focus on the formation of organic nitrates. Strong evidence that both the RH and different $NO_x$ levels have a significant effect on SOA formation from furan photooxidation are presented.

## 2 Experiments

### 2.1 SOA sample preparation

All experiments were performed in a $1.3\,\text{m}^3$ Teflon film chamber at the Institute of Atmospheric Physics, Chinese Academy of Sciences, Beijing. To maximize and homogenize the interior light intensity, a mirror surface stainless steel was chosen as the interior wall of the enclosure. A total of 42 black lamps (F40BLB, GE, Fairfield, CT, USA) with an emission band centre at 365 nm were equipped in the facilities to simulate the spectrum of the UV band in solar irradiation. The chamber was equipped with multiple sampling ports, which allowed the introduction of clean air, seed aerosols, and gas-phase reagents and for measurements of both gas-phase and particle-phase compositions. The reaction set-up used in the present study and the schematic of the smog chamber facility are shown in Fig. S1 in the Supplement (Liu et al., 2017; Jia and Xu, 2014, 2016; Ge et al., 2017a, b).

The zero air (generated from Zero Air Supply, Model 111 and Model 1150, Thermo Scientific) was used as the background gas for the experiments, and three big hydrocarbon traps (BHT-4, Agilent, Santa Clara, CA, USA) coupled with three activated-carbon filters were used to get further purified air. The chamber was cleaned by flowing pure air for at least 4 h prior to each experiment until residual NO, $NO_2$, $O_3$, or any other particles could not be detected in the chamber. The prepared gas-phase furan was introduced into the chamber directly by a syringe. A certain amount of $NO_2$ was injected into the chamber by a gas-tight syringe to achieve the initial $NO_x$ level. For the enhancement of the SOA formation, NaCl seed aerosols were injected into the chamber via the atomization of NaCl aqueous solution with a constant-rate atomizer (Model 3076, TSI, USA). An aerosol neutralizer (Model 3087, TSI, USA) was used to bring particles to a steady-state charge distribution before they were introduced into the reactor. The initial seed number and mass concentrations were approximately $5 \times 10^4\,\text{cm}^{-3}$ and $6\,\mu\text{g\,m}^{-3}$ on average, respectively. For different RH conditions, the dry zero air was introduced into a bottle of high-purity water to control the humidity of the background air. The humidity in the chamber was detected with a hygrometer (Model 645, Testo AG, Lenzkirch, Germany). The initial experimental conditions considered in this study are summarized in Table 1. Typically, approximate 750 ppb furan was employed. To study the effect of $NO_x$ levels on SOA formation, experiments were conducted with initial furan / $NO_x$ ratios ranging from 7.8 to 48.1, whereas to develop and test the role of RH, experiments were performed for RH values varying from 5 % to 85 % and with an initial average furan to $NO_x$ ratio of 7.55. The temperature of the chamber was controlled to be 308–310 K during all the experiments.

Particles and gas-phase species may get lost to the chamber walls on short timescales, thereby influencing the gas-phase chemistry and SOA formation. The wall loss rate constants for $O_3$, $NO_x$, and aerosol particles were $3.3 \times 10^{-7}$, $4.1 \times 10^{-7}$, and $3.6 \times 10^{-5}\,\text{s}^{-1}$, respectively, which were detected from our previous study conducted in the same set-up and similar experimental conditions (Ge et al., 2017a). However, no wall loss of furan was observed within the uncertainty of the detection of the instrument. Furthermore, the light intensity in the reactor was determined to be $0.34\,\text{min}^{-1}$ by using the $NO_2$ photolysis rate constant as the indirect representation. Before the start of each experiment, the static electricity of the Teflon chamber was removed. After all the reactants flowed into the chamber, the reactor was maintained in the dark for at least 40 min without any activities to reach homogeneous mixing.

### 2.2 Gas and particle monitoring

The concentrations of $O_3$ and $NO_x$ were measured continuously by a UV photometric $O_3$ analyser (Model 49C, Thermo Environmental Instruments Inc.) and a chemiluminescence analyser (Model 42C, Thermo Electron Corporation, USA), respectively. The sampling flow rate was $0.75\,\text{L\,min}^{-1}$ for both $O_3$ and $NO_x$ analysers. The uncertainty in the gas-phase measurements was below $\pm 1\,\%$.

**Table 1.** Summary of initial conditions, $O_3$ concentrations, and particle mass concentrations in furan–$NO_x$–NaCl photooxidation experiments

| Exp. no. | Initial conditions | | | | SOA formation results | | | | | | |
|---|---|---|---|---|---|---|---|---|---|---|---|
| | $[\text{furan}]_0$ (ppb) | $[\text{NO}]_x$ (ppb) | RH (%) | $C_4H_4O/NO_x$ (ppbC ppb$^{-1}$) | $O_3$ (ppb) | PM[a] ($\mu$g m$^{-3}$) | NaCl[b] ($\mu$g m$^{-3}$) | NaNO$_3$[c] ($\mu$g m$^{-3}$) | ALW[d] ($\mu$g m$^{-3}$) | SOA[e] ($\mu$g m$^{-3}$) | SOA yield (%) |
| 1 | 708.4 | 16.8 | < 5 % | 48.1 | 91 | 12.2 | 11.3 | 3.5 | – | 1.0 ± 0.1 | 0.04 ± 0.01 |
| 2 | 749.0 | 23.2 | < 5 % | 36.6 | 115 | 15.4 | 10.7 | 1.6 | – | 1.2 ± 0.2 | 0.05 ± 0.01 |
| 3 | 752.5 | 44.7 | < 5 % | 16.9 | 197 | 21.3 | 14.1 | 2.1 | – | 5.1 ± 0.5 | 0.3 ± 0.02 |
| 4 | 705.8 | 51.8 | < 5 % | 13.6 | 250 | 23.5 | 12.6 | 2.5 | – | 8.4 ± 0.9 | 0.3 ± 0.03 |
| 5 | 783.4 | 94.9 | < 5 % | 8.2 | 372 | 29.1 | 13.5 | 3.9 | – | 12.2 ± 1.3 | 0.5 ± 0.05 |
| 6 | 763.4 | 97.5 | < 5 % | 7.8 | 382 | 38.6 | 11.5 | 3.5 | – | 23.5 ± 2.3 | 1.0 ± 0.1 |
| 7 | 764.8 | 96.7 | 23 % | 7.9 | 359 | 55.5 | 10.4 | 3.8 | – | 42.3 ± 4.2 | 1.9 ± 0.2 |
| 8 | 740.1 | 97.3 | 37 % | 7.6 | 353 | 64.2 | 11.2 | 4.5 | – | 48.6 ± 4.9 | 2.2 ± 0.3 |
| 9 | 719.0 | 100.2 | 42 % | 7.2 | 329 | 111.1 | 9.4 | 5.2 | – | 96.4 ± 9.7 | 4.5 ± 0.5 |
| 10 | 704.8 | 92.9 | 54 % | 7.6 | 280 | 138.7 | 12.0 | 6.2 | 20.1 | 100.2 ± 10.1 | 4.7 ± 0.4 |
| 11 | 699.3 | 102.9 | 80 % | 6.8 | 253 | 144.1 | 8.2 | 7.6 | 25.3 | 103.0 ± 10.3 | 4.8 ± 0.5 |
| 12 | 780.7 | 98.1 | 85 % | 8.0 | 241 | 173.0 | 10.0 | 11.1 | 32.6 | 119.2 ± 10.4 | 5.0 ± 0.5 |

[a] PM: particle mass concentration in the chamber was determined from the SMPS and was the sum of NaCl, NaNO$_3$, ALW, and SOA at the end of the experiments.
[b] NaCl: the amount of NaCl at the end of the experiments. [c] NaNO$_3$: the amount of NaNO$_3$ at the end of the experiments. [d] ALW: the amount of aerosol liquid water content at the end of the experiments. [e] SOA: the amount of secondary organic aerosol at the end of the experiments.

The experiment samples were collected by Tenax absorption tubes (150 mm length × 6 mm O.D., 0.2 g sorbent). Furan concentrations and product samples were detected by thermal-desorption gas chromatography mass spectrometry (TD-GC-MS): the thermal desorber (Master TD, Dani, Italy) was combined with gas chromatography (Model 7890A, Agilent Tech., USA) interfaced to a mass selective detector (5975C, Agilent Tech., USA). The initial reactant concentrations are given in Table 1. The chromatographic analytical method used for the analysis was as follows: the chromatographic column was kept at 338 K for 4 min, then heated to 598 K at a heating rate of 20 K min$^{-1}$, and held for 5 min.

The particle size distribution and mass concentration were measured with a scanning mobility particle sizer (SMPS, Model 3936, TSI, USA) composed of a TSI 3081 differential mobility analyser (DMA) and a TSI 3776 condensation particle counter (CPC). The sampling and sheath flow rates of the SMPS were 0.3 and 3 L min$^{-1}$, respectively. Given this flow conditions of the SMPS, particle sizes (in counts cm$^{-3}$) were recorded in 64 size bins for particles in the 13.6 to 710.5 nm size range. The accuracy of the particle number concentration is ±10 % at < 3 × 10$^5$ cm$^{-3}$. The method of the reduced dry ambient size spectrometer (DASS) was used to detect the ALW (Engelhart et al., 2011). To get dry particle mass concentrations, a large diameter Nafion dryer (Permapure MD-700-48F-3) and a multi-tube Nafion dryer (Permapure PD-200T-24E-M) were added to the sampling inlet and sheath flow, respectively. This DASS method is generally accepted and widely used in the detection of ALW content, and can remove up to 90 % of the water vapour without losing the organic-bound water. It should be noted that the dissolved water-soluble species would evaporate back into the gas phase during the ALW measurement when the aerosol water is removed. In fact, the repartitioning of water-soluble components between gas and particle phases was not taken into consideration. The SOA concentrations for high-RH conditions were slightly underestimated, but the underestimation is extremely low and can be negligible. The results were compared with the mass concentrations detected in humid mode, in which the humid air in the chamber was introduced into the SMPS with the sheath air set to 10 L min$^{-1}$. The ALW was calculated as the difference between the particle mass concentrations determined in dry and humid modes (Jia and Xu, 2018). On the basis of the recorded particle volume concentration and assumed particle density, a total suspended particulate (TSP in $\mu$g m$^{-3}$) could be obtained. As shown in Eq. (1), TSP is the sum of seed aerosol (NaCl in this work), NaNO$_3$, particle ALW, and SOA. By dissolving the SOA collected on ZnSe disks into high-purity water, an ion chromatograph (IC, Dionex ICS-900, Thermo Fisher, USA) was then used to analyse the inorganic content (Cl$^-$ and NO$_3^-$) in SOA. With known initial NaCl seed aerosols and their decay rates, the particle mass concentration of NaCl can be calculated, and based on the detected ALW during experiments, the SOA produced in the photooxidation of furan can be estimated from Eq. (1).

$$\text{TSP} = \text{NaCl} + \text{NaNO}_3 + \text{ALW} + \text{SOA} \qquad (1)$$

### 2.3 Chemical characterization of products

The SOA particles were sampled on ungreased ZnSe disk (25 mm in diameter) using a Dekati low-pressure impactor (DLPI, Dekati Ltd., Kangasala, Finland) after the appearance of the $O_3$ maximum concentration. The sampling flow rate of DLPI was 10 L min$^{-1}$ with particle sizes from 30 nm to 10 $\mu$m, classified into 13 stages. According to the particle distribution explored by the SMPS, the SOA particle size was mainly around 150 nm after 3 h of irradiation. The ZnSe disk

was placed on stage 3 to reach the maximum collection of the particles. Afterward, the ZnSe disk was put in the FTIR (Nicolet iS10, Thermos Fisher, USA) sample holder, which had been flushed with $N_2$ to eliminate the impact of ambient $H_2O$ and $CO_2$ on the determination of the chemical composition of formed SOA particles. The spectra were recorded at $4 \, cm^{-1}$ resolution with 128 scans. The samples collected on the ZnSe disk were then dissolved with high-purity water for analysing inorganic species.

To obtain chemical characterization information, the ultrahigh performance liquid chromatograph (UPLC; Ultimate 3000, Thermo Scientific, USA) HESI-Q Exactive-Orbitrap MS (Q Exactive, Thermo Scientific, USA) was used for the analysis of organic compounds produced from the photooxidation of furan. Methanol (Optima™ LC/MSGrade, Fisher Chemical, USA) was used as the eluent in the UPLC system. The acquired mass spectrum of SOA was in the range of 80–1000 Da. As a "soft ionization", HESI can provide molecular formulae for thousands of organic aerosol (OA) substances when coupled with a high-resolution mass analyser. The HESI source was used in both positive and negative ion modes using the optimum method for the characterization of organic compounds. A total volume of 300 L was sampled at a flow rate of $0.2 \, L \, min^{-1}$, and a volume of 20 μL of the extraction was used for the measurement. The addition of acetonitrile allows a lower surface tension of the solution and provides a stable electrospray ionization process (Koch et al., 2005). $N_2$ was used as the sheath and auxiliary gas. The desolvation gas temperature was 320°, and the gas flow was $200 \, μL \, min^{-1}$. The capture voltage was set to 3 kV. The Thermo Scientific Xcalibur software (Thermo Fisher Scientific Inc., USA) was used to analyse the data from HRMS.

## 3 Results and discussion

### 3.1 Gas phase and SMPS results

Experiments were divided into two main groups, which were set to study the role of (i) $NO_x$ and (ii) RH on SOA formation. To keep the variables of each set of conditions consistent, almost identical initial gas-phase conditions were required. The addition of furan and seed aerosol in all the experiments was nearly equal. The first set of experiments (Exp. 1–6, Table 1), designed to investigate the effect of $NO_x$ level on SOA formation, used similar RH conditions with different $NO_x$ concentrations. To further assess the effect of OH produced during the furan–$NO_x$ photooxidation, four experiments (Table S1 in the Supplement, Exp. 13–16) were conducted by injecting $H_2O_2$ into the chamber before the experiments started. For Exp. 6–12, different RH levels coupled with similar furan / $NO_x$ ratios were monitored to assess the RH impact on SOA formation. To analyse the SOA composition, five additional experiments (Table S2, Exp. 17–21) were carried out to analyse the HESI-Q Exactive-Orbitrap

MS, which intended to compare the role of $NO_x$ and RH in the SOA formation. The major gas-phase inorganic chemical reactions which occurred during the experiments are presented in Fig. 1. After turning on the light, the photolysis of $NO_2$ produced NO and O, which further reacted with $O_2$ to form $O_3$ in the chamber. As a result, a slight increase in the beginning followed by a decrease in $NO_x$ concentrations are observed (Fig. 2). On one hand, the decrease in $NO_x$ is caused by the reaction between NO and alkyl $RO_2$, which are formed from the OH-initiated oxidation of furan during the experiment. On the other hand, the oxidation of NO by $O_3$ also contributes to the $NO_x$ decreasing trend.

The photochemical oxidation of VOCs in the presence of $NO_x$ and OH radical is an important source of tropospheric $O_3$ (Labouze et al., 2004). Although the atmospheric degradation mechanisms of furan are complex, the fundamental features of troposphere $O_3$ formation is relatively simple. The main formation pathway of $O_3$ in the atmosphere is the photolysis of $NO_2$ as presented in Fig. 1 (Reactions R1 and R2). The $RO_2$ and $HO_2$ radicals generated from the photooxidation of furan can further react with NO to form $NO_2$ (Reactions R5 and R6), whose photolysis will produce the ground-state oxygen atom (Reaction R1) and thus contribute to the net $O_3$ formation through the reaction of $O_2$ with O (Reaction R2). The very similar VOC level may result in a high $O_3$ production with higher $NO_x$ concentration conditions (Bowman and Seinfeld, 1994). As presented in a previous work on the ethylene–$NO_x$–NaCl irradiations, both experimental results and model simulations showed an abundant $O_3$ production (approximately 500 ppb) with an initial $NO_x$ level of 142 ppb (Jia and Xu, 2016). Therefore, it is important to evaluate the $O_3$ production potential of VOCs in the atmosphere. As shown in Fig. 2, it shows a consistent increasing trend between $O_3$ concentration and SOA mass concentration, indicating that $O_3$ was generated during the furan photooxidation reactions with the formation of SOA. However, it should be noted that the apparent high $O_3$ concentration monitored by the $O_3$ analyser was due to species other than $O_3$ that have an absorption around 254 nm. The generated gas-phase reaction products that have carbonyl, carboxylic acid, and ester groups conjugated with C=C bonds can absorb strongly in the ∼ 200–300 nm range and, accordingly, contribute to the apparent intense signal of $O_3$ detected by the $O_3$ monitor (Strollo and Ziemann, 2013).

The $O_3$ maximum concentration, SOA mass concentration and SOA yield dependences on the $C_4H_4O/NO_x$ ratio are presented in Fig. 2. The results show that the $O_3$ maximum concentration produced during the experiments decreases with increasing $C_4H_4O/NO_x$ ratio. When the $C_4H_4O/NO_x$ ratio decreases from 16.9 to 7.8, there is a sharp increase in $O_3$ concentration from 197 to 382 ppb. Interestingly, experiments for which the $C_4H_4O/NO_x$ ratio changed from 48.1 to 36.60 exhibit a change of 24 ppb in $O_3$ maximum concentration. This result indicates that under different $C_4H_4O/NO_x$ ratio regimes, the concentration of $O_3$ pro-

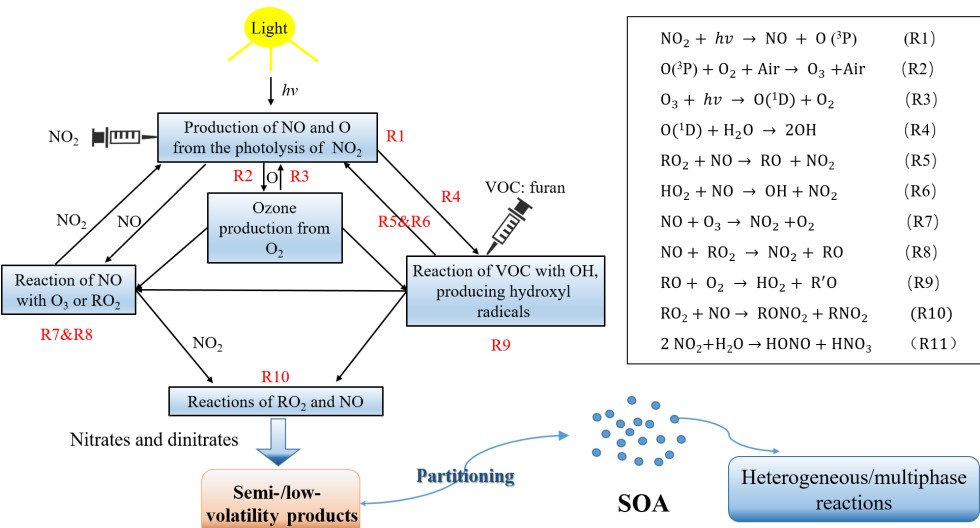

| | |
|---|---|
| $NO_2 + h\nu \rightarrow NO + O\,(^3P)$ | (R1) |
| $O(^3P) + O_2 + Air \rightarrow O_3 + Air$ | (R2) |
| $O_3 + h\nu \rightarrow O(^1D) + O_2$ | (R3) |
| $O(^1D) + H_2O \rightarrow 2OH$ | (R4) |
| $RO_2 + NO \rightarrow RO + NO_2$ | (R5) |
| $HO_2 + NO \rightarrow OH + NO_2$ | (R6) |
| $NO + O_3 \rightarrow NO_2 + O_2$ | (R7) |
| $NO + RO_2 \rightarrow NO_2 + RO$ | (R8) |
| $RO + O_2 \rightarrow HO_2 + R'O$ | (R9) |
| $RO_2 + NO \rightarrow RONO_2 + RNO_2$ | (R10) |
| $2\,NO_2 + H_2O \rightarrow HONO + HNO_3$ | (R11) |

**Figure 1.** Major chemical reactions during the experiments.

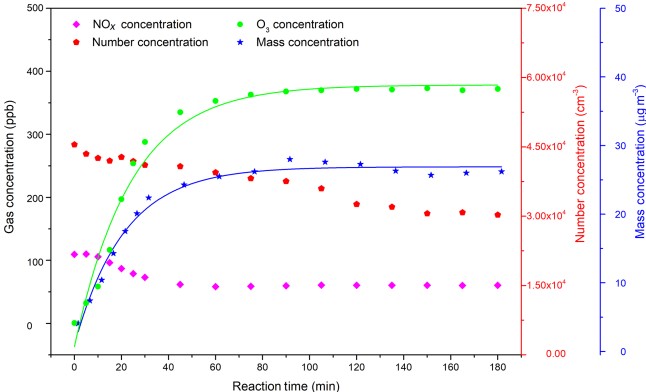

**Figure 2.** Profile of the gas-phase concentrations of reactants (NO, $NO_2$, $NO_x$, and $O_3$) and particle number / mass concentrations (corrected with wall loss) over time. The $C_4H_4O/NO_x$ ratio is 7.9 and RH = 23 %. Since the particle wall loss has a weak RH dependence in our chamber, a mean value of $4.7 \times 10^{-5}\,s^{-1}$ was used for wall loss correction. A density of $1.4\,g\,cm^{-3}$ was used in the SMPS (Jia and Xu, 2018; Kostenidou et al., 2007).

duced varies much. There is a great change in $O_3$ accumulation when the $C_4H_4O/NO_x$ ratios are relatively low (< 16.9), which is consistent with a previous observation from $C_3H_6$–$NO_x$–NaCl irradiation experiments by Ge et al. (2017a), who found that when the $C_3H_6/NO_x$ ratio was less than 11, the $O_3$ concentration decreased considerably with increasing ratio, whereas when the ratio was larger than 11, the $O_3$ concentration slightly decreased with increasing ratio. Figure S2 shows the comparisons of the observed concentrations of the apparent $O_3$ and $NO_x$ from furan irradiations at different experimental conditions. With similar $NO_x$ level, the amount of $O_3$ formed at 5 % RH was larger than that at 42 % RH. The apparent $O_3$ maximum concentration was reduced by almost

60 % as the $C_4H_4O/NO_x$ ratio increased from 7.8 to 36.6. However, the effect of RH on $O_3$ formation is not so obvious compared to the difference in initial $C_4H_4O/NO_x$ ratio. The influence of $NO_x$ level on $O_3$ is likely related to its formation mechanism. The formation of $O_3$ is directly connected with the $NO_x$ in two ways: the photolysis of $NO_2$, producing the O atom that reacts with $O_2$, and the $NO_x$ reacting directly with RO. However, the RH has little to no effect on $O_3$ formation. This formation is slightly favoured at low RH, whereas, at high RH, the $ONO_2$-containing compounds are easily transferred into the aerosol phase, thereby suppressing the $O_3$ formation (Jia and Xu, 2014). Additionally, the slight change in $O_3$ maximum concentration under different RH conditions may also be caused by the consumption of gas-phase reaction products that contain functional groups conjugated with C=C bonds and respond to the $O_3$ analyser. This assumption is reasonable because these carbonyl-rich products were favourable to condense on the moist surface of particles and thus lowered the $O_3$ concentration detected by the $O_3$ analyser, which has been further confirmed by the MS results with enhanced intensities of corresponding organonitrates. The wall loss of the organonitrates species would also explain the decreasing trend of $O_3$ concentration as RH increases. The appearance time of $O_3$ maximum concentration at a high $C_4H_4O/NO_x$ ratio is almost half an hour earlier than that at a low initial $C_4H_4O/NO_x$ ratio. By contrast, there is no significant difference in the balance time of $NO_x$ concentration under different experimental conditions. Similar initial $C_4H_4O/NO_x$ ratios result in a similar profile of $NO_x$ concentration during the experiment, although the final $NO_x$ concentration shows a slight difference.

The calculated SOA yield is defined as the ratio of the mass concentrations of the maximum SOA formed ($M_{SOA}$ in $\mu g\,m^{-3}$, corrected by wall loss) and the reacted furan

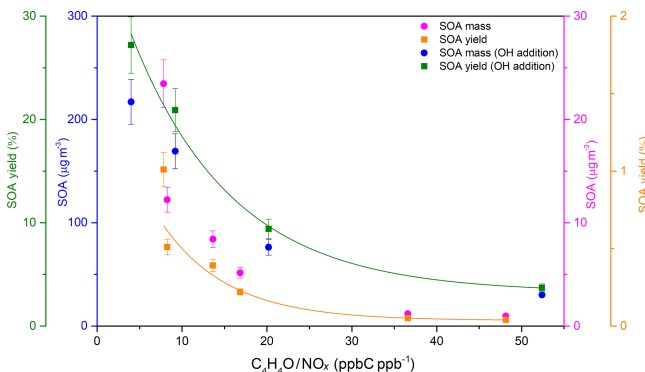

**Figure 3.** Dependence of the SOA mass concentration and SOA yield on the $C_4H_4O/NO_x$ ratio. Since the particle wall loss has a weak RH dependence in our chamber, a mean value of $3.6 \times 10^{-5}\,s^{-1}$ was used for wall loss correction. A density of $1.4\,g\,cm^{-3}$ was used in SMPS (Jia and Xu, 2018; Kostenidou et al., 2007).

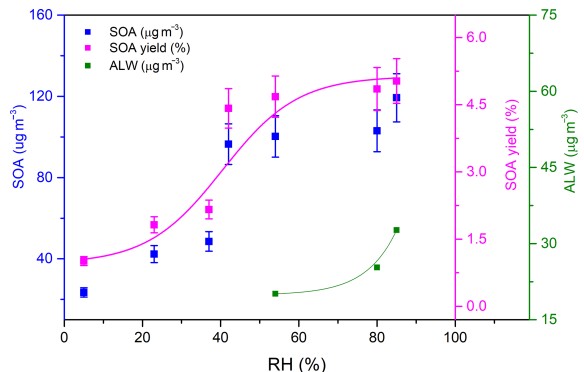

**Figure 4.** Dependences of the SOA mass concentration, SOA yield, and ALW on relative humidity (RH).

($\Delta C_4H_4O$ in $\mu g\,m^{-3}$):

$$Y_{SOA} = (M_{SOA}/\Delta C_4H_4O) \times 100\,\%, \qquad (2)$$

where $M_{SOA}$ is obtained by subtracting the amount of NaCl and $NaNO_3$ from the total particle mass concentration measured by the SMPS and $\Delta C_4H_4O$ is the consumption of furan during the experiments. Experiments were started with an average of $6\,\mu g\,m^{-3}$ of NaCl seed aerosol and were conducted under different initial $C_4H_4O/NO_x$ ratios. The dependences of the SOA mass concentration and SOA yield on the $C_4H_4O/NO_x$ ratios and RH conditions are shown in Figs. 3 and 4, respectively. The numerical values of the aerosol mass concentration and SOA yield are given in Table 1. After the photooxidation reactions, the SOA mass concentrations reached maxima between 1.0 and $23.5\,\mu g\,m^{-3}$, with SOA yield ranging from $0.04\,\%$ to $1.01\,\%$. In other experiments carried out under varying RH conditions, SOA maxima were in the range of $23.5$–$119.2\,\mu g\,m^{-3}$, with SOA yield ranging from $1.01\,\%$ to $5.03\,\%$.

It is generally accepted that experiments with low $NO_x$ levels lead to higher SOA yields than those with higher $NO_x$ levels at the same VOC concentration (Song et al., 2005). However, as shown in Fig. 3, an increasing SOA mass concentration and SOA yield with increasing $NO_x$ was observed. There are two possible explanations for this phenomenon: (i) the concentration of OH radicals produced in situ in the present study before an additional source of OH was insufficient to produce a considerable amount of SOA under low-$NO_x$ conditions. As shown in Fig. S3, the OH concentration exhibits a gradual increase with $NO_x$ concentration and there appears to be a correlation between $NO_x$ concentration, OH concentration and SOA yield. Therefore, at low-$NO_x$ conditions, the increase in SOA yield was attributed to an increase in OH concentration, which was affected by OH recycling following Reaction (R6) (see Fig. 1) and contributed to the enhancement of SOA formation. This result is consistent with a previous study concerning the impact of $NO_x$ and OH on SOA formation from $\beta$-pinene photooxidation, which has proved that the positive correlation between SOA yield and $NO_x$ levels ($[VOC]_0/[NO_x]_0 > 10\,ppbC\,ppb^{-1}$) was caused by the $NO_x$-induced increase in OH concentration (Sarrafzadeh et al., 2016). (ii) Differently, Sarrafzadeh et al. found that after eliminating the effect of OH concentration on SOA mass growth, SOA yield only decreased with increasing $NO_x$ levels (Sarrafzadeh et al., 2016). To further investigate the $NO_x$ effect on furan-generated SOA formation under adequate OH conditions, four more experiments (see Table S1) were carried out with an additional injection of $H_2O_2$ as the OH radical source before the start of each experiment. The SOA yield trend at different $C_4H_4O/NO_x$ ratios is also shown in Fig. 3. The continuous growth trend of SOA yield with increasing $NO_x$ concentration at a relatively high $NO_x$ level may result from the partitioning of generated semi- or low-volatility compounds (multifunctional nitrates and dinitrates) into the particle phase, leading to significant furan SOA formation under high-$NO_x$ conditions. Similarly, SOA from OH-initiated isoprene oxidation under high-$NO_x$ conditions was comprehensively investigated by Schwantes et al. (2019), who suggested that low-volatility hydroxyl nitrates and dihydroxyl dinitrates generated conspicuously more aerosol than previously thought (Schwantes et al., 2019). Our results showing the increase in SOA mass formation by high-$NO_x$ conditions also agree with a previous study, which indicated that a high level of $NO_2$ can participate in the OH-induced reaction of guaiacol, consequently leading to the formation of organic nitrates and the enhancement of guaiacol SOA formation (C. Liu et al., 2019).

It is worth noting that under high-RH conditions, as shown in Fig. 1, the $NO_2$ hydrolysis (Reaction R11) can generate nitrous acid (HONO), which has been regarded as a major source of OH. As indicated in Fig. 4, the SOA yields obtained in the present work clearly show a gradual increase with RH. Also shown in Fig. S3 is the dependence of OH and furan concentrations on RH during the experiments de-

termined from the decay of furan using a reaction rate coefficient of $k(\mathrm{OH} + \mathrm{furan}) = 4.01 \times 10^{-11}\,\mathrm{cm}^3\,\mathrm{molecules}^{-1}\,\mathrm{s}^{-1}$ (Atkinson et al., 1983). It is therefore probable that the increase in RH results in high levels of HONO formation in the chamber, which leads to an increase in OH concentration, a faster furan decay rate, and higher aerosol mass yields. This result is in reasonably good agreement with previous studies, which proposed that the amount of products that can partition into the particle phase increases with the increasing rate of hydrocarbon oxidation (Healy et al., 2009; Chan et al., 2007). Moreover, the increasing RH might also enhance the SOA formation due to the fact that the functionalized gasphase components were more favoured to condense on the surface of wet particles (S. Liu et al., 2019).

## 3.2 SOA chemical composition

To get detailed information on the functional groups in SOA formed during the photooxidation of furan, the collected particles were measured by FTIR, which has been proven to be an ideal technique for the detection of functional groups and bond information in aerosol samples. The FTIR spectra of particles collected from furan–$\mathrm{NO}_x$–NaCl with different values of $\mathrm{C}_4\mathrm{H}_4\mathrm{O}/\mathrm{NO}_x$ ratio are shown in Fig. 5. The absorptions of organic functional groups were detected, which further confirmed the SOA formation from the photooxidation of furan. The assignment of the FTIR absorption frequencies is summarized in Table 2. The organic nitrate exhibits typical NO symmetric stretching at $868\,\mathrm{cm}^{-1}$, $\mathrm{NO}_2$ symmetric stretching at around $1341\,\mathrm{cm}^{-1}$, and $\mathrm{NO}_2$ asymmetric stretching at $1614\,\mathrm{cm}^{-1}$ (Jia and Xu, 2016). The absorption at $1067\,\mathrm{cm}^{-1}$ matches the C-O stretching vibration in C-O-C, while the sharp absorption at $1724\,\mathrm{cm}^{-1}$ is the C=O stretching vibration in carboxylic acid and ketones (Sakamoto et al., 2013). The carbon skeleton corresponds to the vibrations between 2850 and $3100\,\mathrm{cm}^{-1}$, where the $\nu(\mathrm{C\text{-}H})$ stretching vibration can be found. Absorptions in the ranges 2850–3000 and $3000$–$3100\,\mathrm{cm}^{-1}$ represent the C-H stretching vibration in saturated carbon ring and unsaturated alkenes, respectively. Correspondingly, the C-H bending vibrations are represented by the absorption between 1350 and $1515\,\mathrm{cm}^{-1}$. The strong broad vibrations at $2400$–$3500\,\mathrm{cm}^{-1}$ are interpreted as the O-H stretching vibration in carboxyl groups and hydroxyl groups (Ge et al., 2017a).

Figure 5 illustrates the influence of $\mathrm{NO}_x$ level on the SOA components during the photooxidation of furan. For each of the functional groups, experiments with the most pronounced SOA formation are those with the lowest initial $\mathrm{C}_4\mathrm{H}_4\mathrm{O}/\mathrm{NO}_x$ ratio. This is further confirmed by comparing the SOA yield for different $\mathrm{NO}_x$ level conditions under approximately similar gas-phase and initial seed conditions. The calculated variations of the relative abundance of FTIR functional groups at different $\mathrm{NO}_x$ levels are presented in Fig. S4, in which the absolute abundances are normalized with respect to the corresponding functional abun-

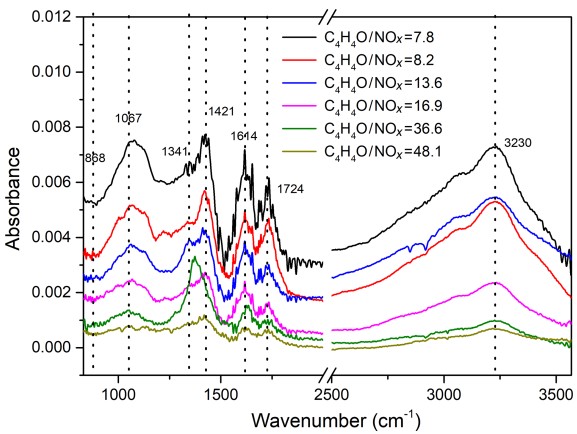

**Figure 5.** FTIR spectra of particles collected from furan–$\mathrm{NO}_x$–NaCl photooxidation experiments with different values of $\mathrm{C}_4\mathrm{H}_4\mathrm{O}/\mathrm{NO}_x$ ratio, ranging from 7.3 to 48.1.

dance detected at $\mathrm{C}_4\mathrm{H}_4\mathrm{O}/\mathrm{NO}_x = 7.8$. The carbonyl compound formed with a 7.8 initial $\mathrm{C}_4\mathrm{H}_4\mathrm{O}/\mathrm{NO}_x$ ratio is approximately three times more abundant than that formed with a 48.1 initial $\mathrm{C}_4\mathrm{H}_4\mathrm{O}/\mathrm{NO}_x$ ratio. The absorbance of $\mathrm{NO}_2$ functional groups exhibits a much stronger enhancement under an initial $\mathrm{C}_4\mathrm{H}_4\mathrm{O}/\mathrm{NO}_x$ ratio of 7.8 compared to 48.1. Although at different $\mathrm{C}_4\mathrm{H}_4\mathrm{O}/\mathrm{NO}_x$ ratio conditions the intensities of C-O-C and O-H functional groups show similar trends, their variations are substantially different from the variations of other functional groups. In sum, the increased absorbance of functional groups with decreasing initial $\mathrm{C}_4\mathrm{H}_4\mathrm{O}/\mathrm{NO}_x$ ratios demonstrates that a relatively high $\mathrm{NO}_x$ level contributes to the formation of SOA.

The FTIR results in the investigation of furan photooxidation under different RH conditions are shown in Fig. 6. The absorbance of the FTIR characteristic peak increases with the RH rising from 5 % to 85 %. Note that obvious intensities of the functional groups were observed when the RH exceeded the efflorescence RH of NaCl. In contrast, the absorptions of corresponding functional groups enhance gently when the RH is lower than 42 %. This phenomenon is consistent with the results of the SOA yield discussed above and can also be interpreted as the increasing ALW components contributing to SOA formation. The ratio of the absorbance intensities detected at low RH to that at 85 % RH is used as the relative abundance to show more intuitive FTIR results. As shown in Fig. S5, for RH between 5 % and 37 %, the intensities of all functional groups vary weakly and are approximately one-third of the intensity at 85 % RH. However, when the RH rises from 42 % to 85 %, the absorption intensities of O-H and $\mathrm{NO}_2$ functional groups increase by factors of 2.0 and 1.9, respectively. In this RH range, the variations of relative intensities are even stronger for C=O, C-H, and C-O-C, being increased by factors of 2.3, 2.3, and 2.5, respectively. A previous study has observed that in an urban environment containing aromatic hydrocarbons and $\mathrm{NO}_x$, the

**Table 2.** Assignment of the observed FTIR absorption frequencies ($cm^{-1}$).

| Absorption frequencies | Functional group | Assignment |
|---|---|---|
| 1222–868 | C-C and C-O | Stretching in alcohols (Coury and Dillner, 2008) |
| | | C-O of COOH group (Duarte et al., 2005) |
| 1067 | C-O-C | C-O stretching (Jang et al., 2002) |
| 1515–1350 | -CH | Bending vibration of -C-H group |
| 868 | -NO | NO symmetric stretching (Jia and Xu, 2016) |
| 1614, 1341 | -$ONO_2$ | $NO_2$3symmetric stretching and asymmetric stretching |
| 1724 | C=O | Stretching vibration in carboxylic acid and ketones |
| 3000–2850 | C-H | C-H stretching vibration in the saturated carbon ring |
| 3100–3000 | C-H | C-H stretching vibration in unsaturated alkene |
| 3200–2400 | O-H | Stretching vibration in carboxyl groups |
| 3100–3500 | O-H | Stretching vibration in hydroxyl groups |

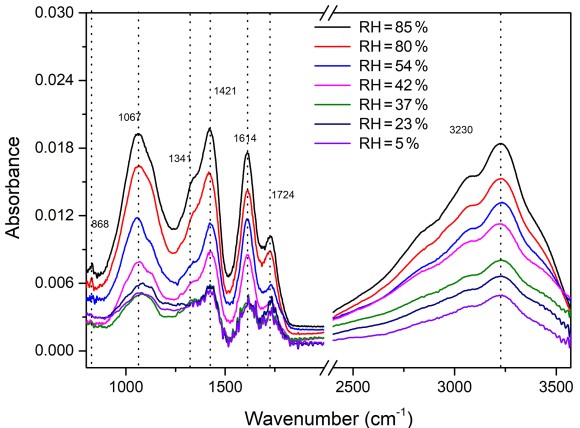

**Figure 6.** FTIR spectra of particles collected from furan–$NO_x$–NaCl photooxidation experiments with different values of RH, ranging from 5 % to 85 %.

SOA yield increased by a factor of 2 to 3 under high RH compared to lower RH (Zhou et al., 2011). This trend is similar to that found in experiments with toluene in an urban aromatic hydrocarbon–$NO_x$ mixture system (Kamens et al., 2011). It should be pointed out that observed FTIR results show a high degree of consistency with the SOA mass concentration and SOA yield.

To further identify and confirm the structure of generated SOA components, more techniques, such as GC/MS and HESI-Q Exactive-Orbitrap MS, were used to analyse the chemical composition. The chromatography before electrospray ionization (ESI)-MS analyses is helpful to remove the disturbances of the inorganic salts and determine the appropriate molecular compound (Surratt et al., 2007; Gao et al., 2006). However, the collected PM products were not sufficient for the chromatography owing to the limitation of the chamber volume. Furthermore, the low SOA yield of the furan photooxidation make it harder to retain enough SOA components for the better response of ESI-MS signal. Consequently, direct infusion analyses were carried out for HESI-

Q Exactive-Orbitrap MS in the present study. Five experiments of furan–$NO_x$–NaCl photooxidation that were conducted under different initial $C_4H_4O/NO_x$ ratios and RH conditions were analysed by HESI-Q Exactive-Orbitrap MS. The mass spectra recorded in different ion modes represent the detected compounds ionizable in either positive or negative modes (Walser et al., 2008). The MS spectra of generated species from different $NO_x$ level and RH conditions, which show evidence for the OH-furan reaction, are presented in Figs. 7 and 8, respectively. The major peaks are $m/z^+$ 85.0018, 101.0894, and 185.0504 in the positive ion mode and $m/z^-$ 146.0161, 225.0125, and 263.0132 in the negative ion mode. The prominent peaks in the HR-MS spectra detected in negative ion mode are comprised of various functionalized hydroxyl nitrates and dihydroxyl dinitrates. However, in the positive ion mode analysis, most carbonyl components were detected. The assignments of these ion peaks, the molecular weights of the products observed, and proposed structures are summarized in Table 3. These detected compounds provide additional evidence for the proposed radical reaction mechanism.

According to the identified products in this work and based on previous kinetic (Atkinson et al., 1983; Lee and Tang, 1982) and product (Villanueva et al., 2009, 2007; Aschmann et al., 2014; Tapia et al., 2011; Strollo and Ziemann, 2013) studies, a proposed chemical mechanism for SOA formation from furan in the presence of $NO_x$ is shown in Scheme 1. Additionally, on the basis of well-established mechanisms for atmospheric volatile organic compounds (Atkinson and Arey, 2003), it can be concluded that the reaction is initiated by OH addition to a C=C bond at C2 or C3 positions. Addition at the C2 position forms two cyclic alkyl radicals (a, c), one of which (c) can isomerize to form a ring-opened alkyl radical (d), whereas addition at the C3 position forms a single alkyl radical (b). The OH radical addition leading to a hydrogen abstraction generates the alkyl radicals (R•) followed by reaction with $O_2$ to form alkylperoxy radicals ($RO_2$•) (Pan and Wang, 2014). Moreover, the formed alkylperoxy radicals ($RO_2$•) can either react with $RO_2/HO_2$ or NO to yield

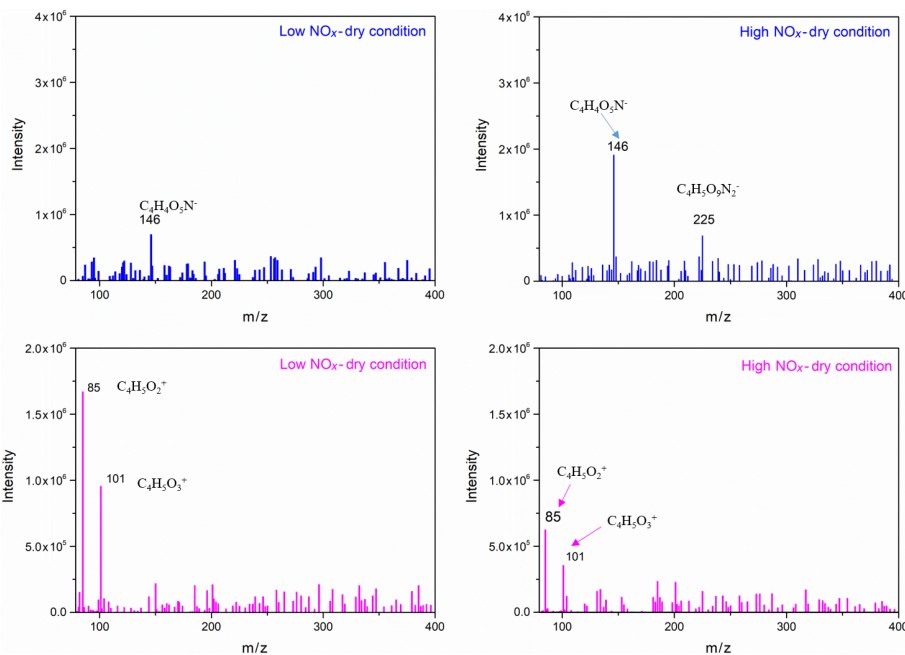

**Figure 7.** Selected background-subtraction HESI-Q Exactive-Orbitrap MS results of SOA in both negative (blue) and positive (pink) ion modes from the photooxidation of furan under different RH conditions.

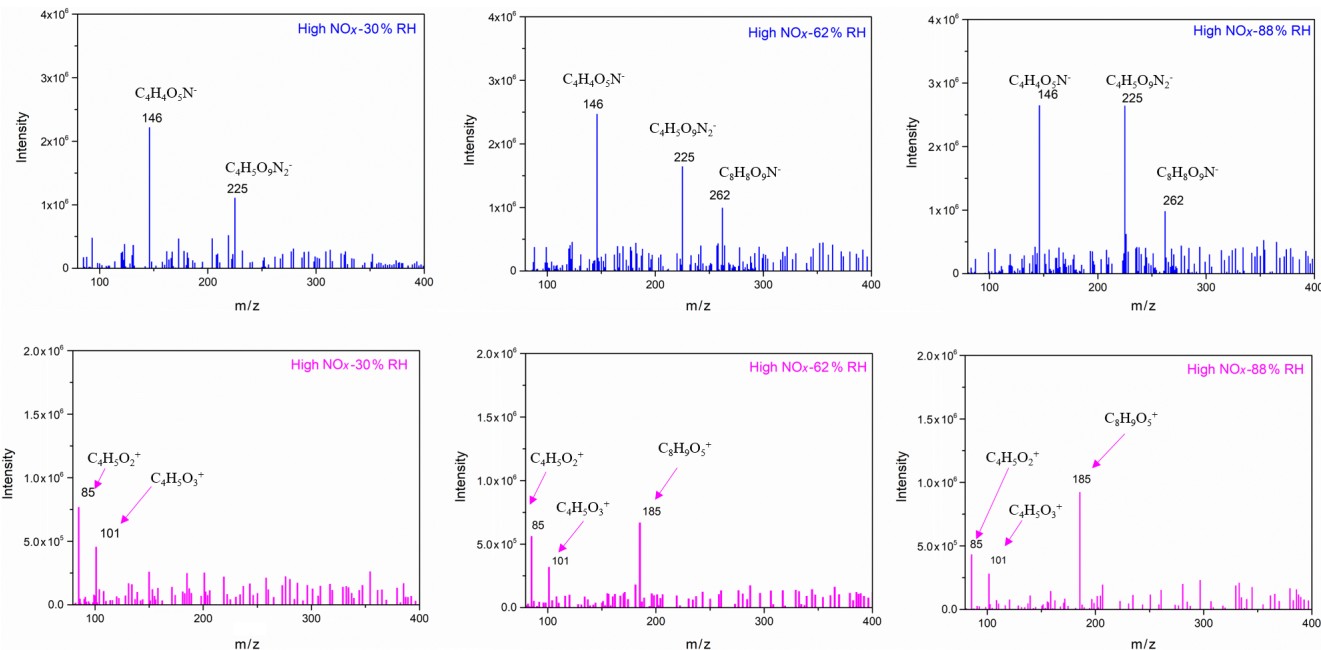

**Figure 8.** Selected background-subtraction HESI-Q Exactive-Orbitrap MS results of SOA in both negative (blue) and positive (pink) ion modes from the photooxidation of furan under different RH conditions.

the corresponding alkoxy radical (RO•), which can (i) decompose and then react with $O_2$ to yield a 1,4-aldoester (A), (ii) react with $NO_2/NO/O_2$ to form hydroxyl nitrate compound isomers with $m/z^-$ 146, or (iii) react with $O_2$ to form unsaturated products and hydroxyfuranone (B) and 1,4-aldoacid (C). The ring-opened alkylperoxy radical generated from (d) can decompose to generate an unsaturated 1,4-dialdehyde (D). The formation of 1,4-dialdehyde with $m/z^+$ 85 detected in the positive ion mode suggests that these unsaturated 1,4-dicarbonyls are formed after initial OH addi-

**Table 3.** Ion peaks with the assigned compounds observed in the HESI-Q Exactive-Orbitrap MS. Proposed assignments are based on the formula from HESI-Q Exactive-Orbitrap MS.

| Ion mode | No | Mass ($m/z$) | Ion mode | Ion formula | Delta (amu) | Proposed structure |
|---|---|---|---|---|---|---|
| Positive ion mode | 1 | 85.0018 | $[M+H]^+$ | $C_4H_5O_2^+$ | −0.027 | |
| | 2 | 101.0894 | $[M+H]^+$ | $C_4H_5O_3^+$ | 0.066 | |
| | 3 | 185.0504 | $[M+H]^+$ | $C_8H_9O_5^+$ | 0.006 | |
| Negative ion mode | 4 | 146.0161 | $[M-H]^-$ | $C_4H_4O_5N^-$ | 0.007 | |
| | 5 | 225.0125 | $[M-H]^-$ | $C_4H_5O_9N_2^-$ | 0.012 | |
| | 6 | 262.0132 | $[M-H]^-$ | $C_8H_8O_9N^-$ | −0.007 | |

tion at 2- or 5-positions. We note that OH radical addition at 2, 3-positions would lead to carbonyl product isomers with the same $m/z^+$ 101. In addition, some dihydroxyl dinitrates with $m/z^-$ 225 were also detected in negative ion mode. However, the pathways favouring the generation of these dihydroxyl dinitrates could only take place under high $NO_x$ levels. Scheme 1 shows the formation of second-generation products hemiacetals (E) via the reactions of the hydroxyfuranone (B) with 1,4-dialdehyde (D). After uptake from the gas phase, the combination of hydroxyfuranone with 4-dialdehyde appeared to occur by H abstraction, followed by dehydration, thus forming $m/z^+$ 185 compounds. This reaction pathway has also been identified in previous studies of OH-initiated reactions of furans (Strollo and Ziemann, 2013; Aschmann et al., 2014). According to the results of HR-MS, this aqueous-phase reaction is more favoured in aqueous particles.

## 3.3 Effects of $NO_x$ on SOA formation

To study the $NO_x$ level effect on SOA formation from the photooxidation of furan, the experiments were conducted with varying initial $C_4H_4O/NO_x$ ratios ranging from 7.8 to 48.1. The SOA formation is found to have much lower yield under a high $C_4H_4O/NO_x$ ratio. The SOA mass concentration and SOA yield increased from 1.0 to 23.5 µg m$^{-3}$ and 0.04 % to 1.01 %, respectively, as the initial concentration ratio of $C_4H_4O/NO_x$ decreased from 48.1 to 7.8 (ppbC ppb$^{-1}$). This trend is consistent with previous studies on propylene photooxidation, which found that the SOA yield was enhanced under a low VOC/$NO_x$ ratio (Ge et al., 2017a). Another laboratory study of aerosol assessment from isoprene photooxidation concluded that under a low $NO_x$ level (< 129 ppb), the SOA mass increased with increasing initial $NO_x$ level (Kroll et al., 2006). By comparing the studies of Ge et al. (2017a) and Kroll et al. (2006), there is a positive correlation between $NO_x$ effect and SOA formation when the initial $NO_x$ level is relatively low (nearly below 100 ppb). These results further support our findings in the experiments of furan photooxidation. As shown in Fig. S6, SOA formed under a low $C_4H_4O/NO_x$ ratio also have a faster increase than those under large $C_4H_4O/NO_x$ ratio conditions. However, the particle number distribution under $C_4H_4O/NO_x =$ 7.8 and $C_4H_4O/NO_x = 36.6$ conditions shows similar profiles.

Generally, the $NO_x$ level has two different effects in the reaction process. Firstly, increasing $NO_x$ concentration will

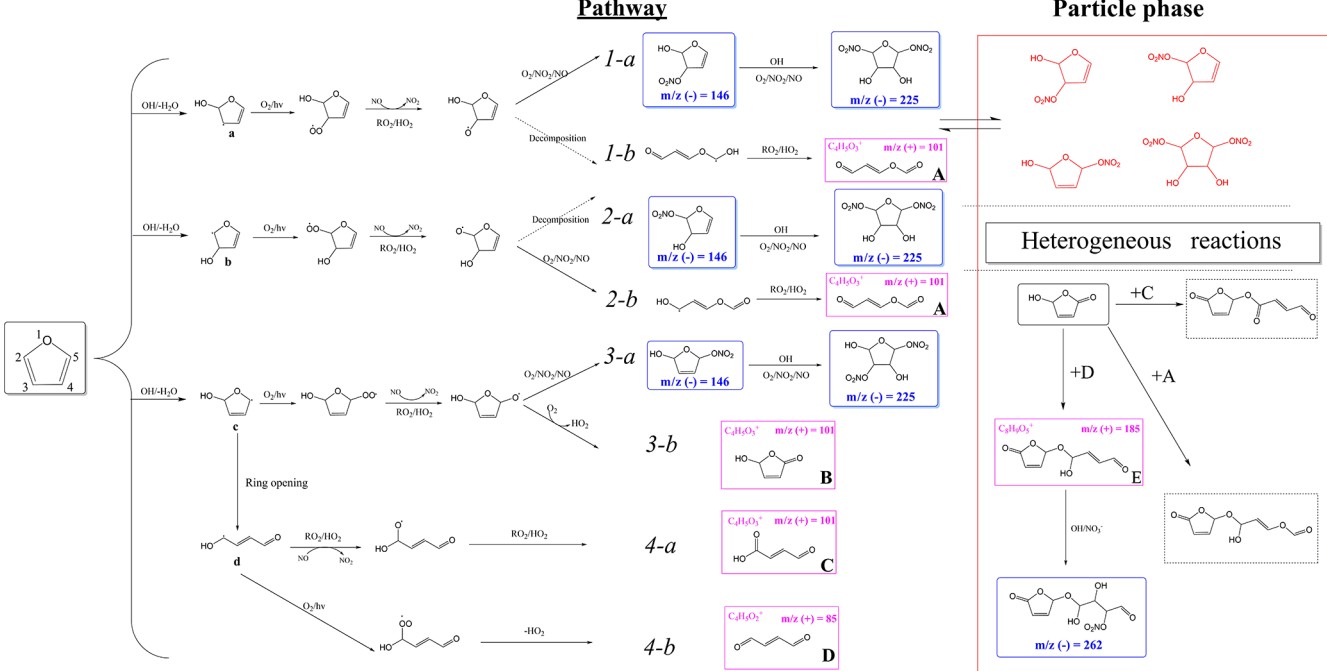

**Scheme 1.** Proposed chemical mechanism of furan–$NO_x$ photooxidation under different experimental conditions. SOA constituents in blue and pink boxes are proposed SOA constituents detected by HESI-Q Exactive-Orbitrap MS under negative and positive ion mode, respectively. The detected nitrates and dinitrates in red are low-volatility organic species, which can easily partition into the particle phase and enhance the SOA formation.

promote the $O_3$ and HONO formation, leading to more OH radical formed, which in turn is favourable to SOA formation (Sarrafzadeh et al., 2016). In addition, sufficient $NO_x$ can facilitate the competition between NO and $HO_2$ to react with $RO_2$. Products with high volatility will be generated more by the $NO+RO_2$ reaction than by the $HO_2+RO_2$ reaction (Kroll and Seinfeld, 2008). However, the formation of lower-volatility products favours the increase in SOA yields (Chen et al., 2018). In this regard, the increasing $NO_x$ level is not conducive to SOA formation. It was shown that the yields of SOA generated from the photooxidation of $m$-xylene increased firstly and then decreased with the increase in the $NO_x$ level (Chen et al., 2018). In the present study, an increasing trend of SOA formation was observed with the increase in $NO_x$ concentration. As shown in the HESI-Q Exactive-Orbitrap MS results, all the detected primary products are carbonyl-rich, and even the organonitrates have at least two carbonyl functional groups. These carbonyl-containing products have lower volatility and contribute to the SOA formation. The peak intensities in the MS of the products ($m/z^+$ 85, 101) generated by the pathways involving $HO_2$ (as indicated by Scheme 1) decreased with the increase in $NO_x$ concentration. Additionally, more products of dihydroxyl dinitrates ($m/z^-$ 225) with multifunctional groups were detected under high-$NO_x$ conditions. As shown in Scheme 1, the multifunctional organonitrates detected in negative ion mode are produced mostly from later-

generation chemistry. Hydroxyl nitrates with $m/z^-$ 146 can be formed through pathways 1-a, 2-a, and 3-a by the reaction of $RO_2$ with $NO_x$. We note that the $m/z^-$ 146 compound was detected both at low $NO_x$ levels and high $NO_x$ levels. However, the peak intensity of this product was decreased with increasing $NO_x$ concentration. This phenomenon might be caused by the later reaction of the unsaturated hydroxyl nitrates going through a second OH-initiated reaction and leading to the formation of the dihydroxyl dinitrate with $m/z^-$ 225. In addition, the peak intensity changes of SOA products detected in the positive mode, such as the peaks at $m/z^+$ 85 and 101, were reduced under high-$NO_x$ conditions, which resulted from the fact that the $RO_2$ radical fate was dominated by the pathway of $RO_2+NO$ or $RO_2+NO_2$. This result supports that the fate of $RO_2$ is not a single channel reaction. There exists a competition between $RO_2$ reacting with $NO_x$ and with $HO_2$ under high-$NO_x$ conditions, but the former pathway is more favourable. There are two pathways for hydroxyl nitrates formation from $RO_2$ radicals in the presence of $NO_x$ according to which $RO_2$ radicals may react with NO and $NO_2$ to form $RONO_2$ and $ROONO_2$, respectively (Kroll and Seinfeld, 2008). However, the formed peroxynitrates could easily thermally dissociate and convert to $RONO_2$.

Furthermore, by analysing the OH concentration and product components, we conclude that there are two possible explanations for the increasing trend of SOA yield as the

$NO_x$ level increases: (i) the SOA production is closely related to the oxidation capacity in the photooxidation experiments. Experiments conducted under different $NO_x$ levels indicate that the OH concentration is controlled by the $NO_x$ level if there is no additional OH precursor added before the start of the experiment. As shown in Fig. S3, an increase in $NO_x$ level results in more OH generation and a faster furan decay rate. This justifies the observed higher SOA mass concentration and higher SOA yield. (ii) HRMS fragments associated with multifunctional organonitrates are enhanced under high-$NO_x$ conditions (Fig. 7). As presented in Scheme 1, the furan dihydroxyl dinitrates are generated from the first-generation hydroxyl nitrate reacting with OH to form a peroxy radical, which reacts thereafter with NO. Together with multifunctional hydroxyl nitrates, these low-volatility species can easily partition into the particle phase and increase the SOA mass concentration. More importantly, the seed particle added initially plays an important role in the processes of gas-particle partitioning as indicated by a recent study, which showed that sufficient seed surface area at the start of the reaction largely suppressed the effects of vapour wall losses of low-volatility compounds (Schwantes et al., 2019). Therefore, the NaCl seed particle added in the present work promoted the partitioning of the formed low-volatility functional organonitrates.

## 3.4 Effect of RH on SOA formation

Experiments 6–12 were conducted under seven different RH conditions ranging from 5 % to 85 %. In this RH range, the SOA yield increases from 1.01 % to 5.03 %. With almost identical initial conditions except RH, the yield of furan-derived SOA formed at high RH can be a factor of 2 higher than that formed at low RH. A similar trend was also observed by Yu et al. (2011), who found that the SOA mass concentrations increased by a factor of 6 when RH increased from 18 % to 82 % (Yu et al., 2011). As shown in Fig. S3, an increase in RH leads to higher OH concentration resulting from higher HONO levels generated by the reaction of $NO_2$ with $H_2O$. Previously, using quantum mechanical calculations, Anglada et al. (2011) confirmed that the water component could increase the OH production. The positive correlation between initial water vapour concentration and OH concentration has also been previously observed experimentally (Healy et al., 2009; Tillmann et al., 2010). Additionally, Healy et al. (2009) have also reported that increasing OH concentration promoted the decay of VOC and enhanced SOA formation (Healy et al., 2009). Similarly, in the present work, a faster decay rate of furan was also observed as RH increased, as shown in Fig. S3. It is possible that the faster rate of gas-phase oxidation under higher OH concentrations will lead to the generation of less volatile compounds as presented previously (Chan et al., 2007). A higher OH concentration promotes oxidation reactions, influences the distribu-

tion of organic products, and facilitates the SOA formation (Sarrafzadeh et al., 2016).

An obvious increase in SOA yield was observed when the RH increased from 37 % to 54 %. This phenomenon was mainly caused by the efflorescence transition when the seed particles were coated with SOA. It has been previously shown that SOA formation decreases both the efflorescence RH and deliquescence RH of the seed particles and results in the uptake of water by the particles (Liu et al., 2018; Takahama et al., 2007; Smith et al., 2012). It is highly possible that the NaCl seeds effloresce and deliquesce early after being coated by the newly formed SOA. The effect of efflorescence contributes to the water uptake by the particles, leading to the obvious trend changing of SOA yield. It is noted that with the NaCl seed aerosols serving as nuclei, the ALW was high at high RH. Products with water solubility produced from the photooxidation of furan can dissolve into the ALW of aerosol particles. As a result, ALW in the formed aerosols plays an important role in gas or particle partitioning. As shown in Fig. 4, the ALW was detected when the RH was higher than 54 %, which was based on the deliquescence of NaCl under high-RH conditions. The increase in ALW could partially explain the increase in SOA mass concentration and SOA yield. It is highly probable that the particle surface area increases with an increasing amount of ALW as shown in Fig. S7, which likely promotes the dissolution of semi-volatile matters produced during the experiments. According to the HESI-Q Exactive-Orbitrap MS results shown in Fig. 8, the intensities of multifunctional hydroxyl nitrates and dihydroxyl dinitrate ($m/z^-$ 146 and 225, respectively) exhibited positive correlations with RH. Slight peak intensity increases in $m/z^+$ 85 and 101 products were also observed under high-RH conditions. This phenomenon indicates that the gas-particle phase partitioning of low-volatility compounds was enhanced under high-RH conditions. Furthermore, the increased surface area under high-RH conditions may also be attributed to the condensation of the produced multifunctional compounds.

Another possibility for the increasing trend of SOA yield with the increase in RH might result from the SOA formation through aqueous chemistry in wet aerosols (Grgic et al., 2010; Lim et al., 2010). In these atmospheric processes, alcohols, aldehydes, and ketones formed from the photooxidation of furan in the gas phase can be absorbed into the humid surface of the hygroscopic SOA at high RH. This process further contributes to the formation of low-volatility products on the SOA surface. In addition, the aqueous photochemistry of highly soluble small compounds that partitioned in ALW could produce additional organic compounds and result in a larger SOA yield under high-RH conditions (Faust et al., 2017; Jia and Xu, 2014). The appearance of $m/z^+$ 185 and 262 detected by the HESI-Q Exactive-Orbitrap MS further demonstrated that aqueous-phase reactions indeed took place under high-RH conditions. As shown in Scheme 1, the peak of $m/z^+$ 185 could form by aqueous-phase reaction of

the hydroxyfuranone (B) and 1,4-dialdehyde (D). The formation of hemiacetal (E) has also been detected by a previous study of the OH-initiated reaction of 3-methylfuran in the presence of $NO_x$ (Strollo and Ziemann, 2013). The proposed hemiacetal compound (E) plays a substantial role in the obvious increase in $m/z^+$ 185 product formation under high-RH conditions. A pathway of organonitrate ($m/z^-$ 262) formation in the aqueous particles with the presence of $NO_3^-$ was suggested in the present study based on a previous work, which indicated a radical–radical reaction pathway for organosulfate formation from aqueous OH oxidation of glycolaldehyde in the presence of sulfuric acid (Perri et al., 2010). Increasing the RH also resulted in an overall addition of peak intensities in the negative ion mode, due to the fact that the sample obtained at high RH during the SOA generation had a larger particle surface. Specifically, a relatively stronger intense band of $C_8H_8O_9N^-$ compound ($m/z^- = 262$) was found under a high RH. Consequently, the heterogeneous products in wet seed particles will further contribute to the formation of SOA because higher aerosol liquid water content enables more aqueous-phase reactions.

In conclusion, the reasons for the increasing trend of SOA formation under high-RH conditions may be summarized as following: firstly, higher concentration of OH radical will certainly promote the SOA formation as RH increases. A faster decay rate of furan will also contribute to the formation of products that can partition into the particle phase. In addition, it is possible that the aqueous surface of seed particles provides a new substrate for the photooxidation of furan. Previously, $N_2O_5$ and $HNO_3$ have been proven to be the key products in the VOC–$NO_x$ irradiation experiments (Wang et al., 2016). The moist surface under high-RH conditions is more favourable for the condensation of products with low vapour pressure, leading to the increasing production of SOA formation. The high-RH environment favours the formation of the hemiacetal compounds. Moreover, the effect of RH on SOA formation in furan photooxidation can also be determined by the aqueous photochemistry under high-RH conditions as discussed above. The aqueous-phase reactions at the surface of particles promote the formation of hemiacetal-like products, which likely plays an important role in the process of SOA formation. Previously, the unsaturated first-generation reaction product of 3-methyl furan has also been suggested to undergo acid-catalysed condensed-phase reactions, with SOA yields of up to 15 % (Strollo and Ziemann, 2013). In addition, the reinforced effect of RH on SOA yield was also ascribed from the photooxidation of other aromatic compounds, such as benzene (Ng et al., 2007b), toluene (Hildebrandt et al., 2009; Kamens et al., 2011), and xylene (Zhou et al., 2011).

## 4  Conclusion

The effects of $NO_x$ and RH on SOA formation from the photooxidation of furan in the presence of NaCl seed particles have been investigated in this study. The results demonstrated that the formation of SOA was promoted when the initial VOC/$NO_x$ ratio decreases. The increase in SOA yield at a high VOC/$NO_x$ ratio was caused by the $NO_x$-induced increase in OH concentration. The reason for the promotion of SOA mass at a low furan/$NO_x$ ratio is that more low-volatility products were generated from the furan photooxidation, which contributes to the formation of SOA. Additionally, the increase in RH results in the increase in the mass concentration of the produced SOA. The mechanisms controlling SOA formation may include the gas-phase photooxidation of furan, physical water uptake as RH increases, and the gas-phase reaction of water with the first-generation products and the aqueous chemistry of low-volatility products reacting at the wet surface of NaCl seed particles. Seed aerosols are important for the growth of atmospheric particles and, therefore, affect aerosol–cloud–climate interactions. Organic nitrates, detected with FTIR and ESI-Exactive-Orbitrap MS, were found to be significant in the composition of newly formed particles. A significant amount of carbonyl-rich products were also detected in the SOA products from the photooxidation of furan. A recent study showed that the reactive nitrogen chemistry in aerosol water can be a source of atmospheric sulfate during haze events (Cheng et al., 2016). In addition, the ALW is closely linked with air quality (Malm et al., 1994) and aqueous SOA formation (Sareen et al., 2017). The influence of the ALW component on SOA yield was examined and it was found that increased ALW amounts lead to higher SOA mass concentration and yield, therefore highlighting the importance of the ALW in photooxidation reactions. ALW plays a crucial role in atmospheric physicochemical processes. The current results could also be used to interpret ambient gas-phase measurements and reaction mechanism inference.

*Data availability.* Data are available by contacting the corresponding author.

*Supplement.* The supplement related to this article is available online at: https://doi.org/10.5194/acp-19-1-2019-supplement.

*Author contributions.* LD and XJ conceived and led the studies. XJ, NTT, LJ, SL, and HZ carried out the experiments and analysed the data. XJ, LD, and SL interpreted the results. NTT, LJ, and YX discussed the results and commented on the paper. XJ prepared the paper with contributions from all co-authors.

*Competing interests.* The authors declare that they have no conflict of interest.

*Special issue statement.* This article is part of the special issue "Simulation chambers as tools in atmospheric research (AMT/ACP/GMD inter-journal SI)". It is not associated with a conference.

*Acknowledgements.* We thank Hartmut Herrmann for his very constructive comments at the revision stage.

*Financial support.* This research has been supported by the National Natural Science Foundation of China (grant nos. 91644214 and 41375129), the Shandong Natural Science Foundation for Distinguished Young Scholars (grant no. JQ201705), the Shandong Key R&D Program (grant no. 2018GSF117040), and the Fundamental Research Funds of Shandong University (grant no. 2017JQ01).

*Review statement.* This paper was edited by Jason Surratt and reviewed by three anonymous referees.

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

**Remarks from the language copy-editor**

CE1  I am afraid that we cannot remove entire sentences without the editor's approval. Is this deletion absolutely necessary? If so, please provide an explanation that can be forwarded on to the editor by us.