# Peer review of "Secondary organic aerosol formation from photooxidation of furan: effects of NOx and humidity"

_Atmospheric Chemistry and Physics, 2019_

## Referee Comment (RC1) · Anonymous Referee #1 · 10 Apr 2019

This study investigates the formation of secondary organic aerosols from the photooxidation of furan at different NOx and RH levels. SOA yields were measured using NaCl seeds to provide surface area for the partitioning of SOA-forming vapors. The chemical composition, in particular organic functional groups and a selection of molecular products, was characterized by FTIR and ESI-MS. The authors found a strong dependence of the measured SOA concentration, mass yield, and the intensity of individual functionalities on the initial VOC/NOx ratios and RH levels in a series of experiments conducted, and suggested that NOx and RH play an important role in the SOA formation from furan oxidation by altering the chemical pathways, e.g., aqueous phase chemistry, that essentially lead to SOA. This conclusion, however, is heavily drawn from the inadequate data analysis and interpretation, and lacks fundamental understanding of the predominant chemistry that occurs in the chamber experiments performed.

**NOx dependence of SOA yields**

In this study, photolysis of NO2 was used to generate O3, which further undergoes photolysis and reaction with H2O to generate OH radicals. The initial VOC/NOx ratio in the performed experiments ranges from ~7 to ~48. With the presence of hundreds of ppb levels of furan at the beginning of the experiment, furan is not completely oxidized at the end and the measured SOA mass is mostly composed of the very first few generations of oxidation products. As the initial VOC/NOx ratio decreases, more NO2 will be available for the formation of O3 and consequently OH radicals. **The observed 'NOx-effect' here is essentially the OH effect: higher OH levels result in more furan consumed, thus producing more SOA mass and higher SOA yield. This OH effect on SOA production has been well studied and understood in the community.**

**RH dependence of SOA yields**

Again, as H2O is used to generate OH radicals, higher RH levels result in more OH radicals, which lead to more SOA mass produced from furan photooxidation. **The observed 'RH effect' is essentially another 'OH effect' by promoting the generation of OH radicals and accelerating the oxidation processes of furan.** The authors suggest that relative humidity affects the SOA yield through aqueous phase chemistry. However, the deliquescence point of sodium chloride is around 70% RH, below which the water content in the NaCl particles is close to zero, meaning that there would be minimal aqueous phase chemistry occurring in the particle phase.

**SOA measurements at high RH levels**

It is well known that using DMA to measure aerosol size distribution and mass leadings under high RH is subject to many certainties, e.g., arcing at high voltage

caused by high water content in the aerosols. While the authors used a diffusion drier in front of the DMA inlet, which could certainly minimize the arcing effect that interferes the measurement of big particles, the drying efficiency was not characterized. Have the authors measured the RH of the aerosol flow upon the exit of the diffusion drier? Did aerosols generated under high RH (e.g., 80%) still carry a certain amount of water after drying? Additionally, the authors need to consider how the drying processes affect the repartitioning of water-soluble components between gas and particle phases in order to obtain an accurate SOA yield.

Treatment of wall losses

The particle and vapor wall loss rates are chamber specific quantities that depend on a number of different parameters, i.e., the chamber size (volume to surface area ratio), the wall materials, the humidity in the enclosure air that affects the static charges on the chamber wall surface, and the mixing conditions (static or active mixing), among many others. The interaction patterns of particles with the chamber walls have been well studied for decades, and the particle wall loss rate has been found to vary substantially, by orders of magnitude, among different chamber environments. That the authors simply took the particle wall loss parameterizations obtained in other chambers to correct their own experiment would no doubt introduce significant uncertainties in their SOA mass measurements, resulting in unreliable SOA yield calculations.

---

## Referee Comment (RC2) · Anonymous Referee #2 · 14 Apr 2019

Comments:
This manuscript aimed to study SOA formation from furan under photooxidation conditions with varied NOx and RH. SOA mass, O3 concentration, and SOA composition were carefully measured. The authors concluded that furan photooxidation is dominated by RO2 + NO chemistry that leads to formation of carbonyl-rich products. SOA formation was found to enhance under higher relative humidity and higher VOC/NOx concentration. The results are clearly presented, but there are a few important issues that need to be addressed before becoming publishable. My major comment is that the manuscript did not provide sufficient chemistry insights given the suite of instruments used.

Major:
1. The authors claims that the O3 maximum concentration decreases with increasing RH likely due to that high RH favors the partitioning of the NOx reservoir, RNO2 and RONO2 into the particle phase. Thus the release of NOx is limited. However, it is unclear how much more pronounced is the RNO2 and RONO2 uptake at higher RH and whether this enhancement is sufficient to affect O3 formation. These species need to be very soluble to show distinct partitioning behavior under different RH. What are their possible structures and could the Henry's law constants be estimated? Were any RNO2 or RONO2 species enhanced in the SOA samples under higher RH? From the ESI-MS results, the authors said that the SOA composition is similar between higher RH and low RH. Does this contradict what the authors concluded earlier? You have a max SOA yield of 5% and only a small fraction of that is RNO2 or RONO2. Thus are they important enough to affect gas-phase NOx concentration? Evidence is needed. Alternatively, could wall loss of the RNO2 and RONO2 under high RH better explain the observation?

2. The ESI-MS results presented grouped ions and observed m/z 200-299 as the most abundant group. What does this mean? What are the major chemical formulas observed in this range? I think a lot more discussion of the ESI-MS results could be included here. For example:
(1) Could the authors group all nitrogen-containing species and discuss their presence?
(2) What are the ranges of O:C ratio and oxidation state?
(3) Any variation as the furan/NOx and RH changed?
(4) For soluble species uptaking onto aqueous particles, they will likely oligomerize. Do the authors see oligomers in the ESI-MS?
(5) There is a substantial decrease of large molecules under lower furan/NOx ratio and higher RH in the negative ion mode (but not in the positive ion mode). What does that suggest?
(6) what is the ion in the negative ion mode around m/z 380?
In the current form, there is very little discussion (page 11, line 16-24, page 12, line 2-7, and page 13, line 8-11).

3. The observation of enhanced SOA formation with increased NO is interesting, because most previous work have shown that increase of NO tend to promote RO2+NO chemistry which lead to fragmentation. Why are SOA yields higher with higher NO in the case of furan photooxidation? The authors gave these possibilities in the text, but did not further conclude it based on the mechanism. Is this because furan is a cyclic compound and the fragmentation does not break the

C4 backbone? Or were OH concentrations very different between different furan/NOx experiments (as in the cases for the two referred studies)? Or both?

4. The discussion of the RH effect on SOA using the mass spectral data was very vague. The authors only discussed two ions, m/z 187 and m/z 255. First, it seems the m/z 187 ion is C4H4O4Cl2 from high-resolution fitting. But how is it formed? The authors proposed Cl adduct for the major ions m/z 187, 255, and 281. It is probabaly the case for m/z 255 and 281, since they contain nitrogen and have only one Cl on the ion. But m/z 187 is probably not. The C4H4O4 compound does not exist in Scheme 1. The compound D is C4H4O3. How does it form C4H4O4 in the mechanism? Further, for the m/z 255 ion, the formulas provided in Figure 6 is incorrect. $[C4H4O4]^{35}Cl_2-$ has an exact mass of 185.949; $[C4HO9N2]^{35}Cl-$ has an exact mass of 255.938; $[C8H8O8N]^{35}Cl-$ has an exact mass of 280.994. The authors need to make sure the chemical formula assignments are correct before discussing the mechanism.

Minor and technical comments:
1. Abstract: The abstract should deliver the most important scientific findings. It is unnecessary to describe how the measurements were performed in the abstract (e.g., how particle mass concentration and size distribution were determined). Additionally, "the SOA mass concentration and yield increase with increasing humidity, because higher aerosol liquid water content brings more aqueous phase reactions" does not make sense. More aqueous phase reactions do not mean higher product yield.

2. Second sentence in the introduction puts different categories together. Remove black carbon and brown carbon.

3. Page 2, line 1. Change volatile organic compound to VOC.

4. In the introduction, some statements about cited studies are incorrect. For example, page 2, line 3-4, the Chan et al. (2010) study demonstrated that isoprene SOA formed under high NO2/NO is large, not under high-NOx conditions; the Zhang et al. (2012) as well as the Zhang et al. 2011 studies show different compositions under dry and wet conditions.

5. Page 10, line 10. What is C4H4O4Cl? Does the authors have a prediction of the possible structure? How does the NaCl interacts with the organic components in SOA to form this species? On page 11, line 24, some discussion was made, but it is still unclear how "Cl adduction of SOA product" happens.

6. Page 10, line 13. The carboxylic acids may not generate enough acidity to catalyze reactions as the authors mentioned, unless the authors could provide evidence.

7. Page 12, line 28. The authors said "it is highly probable that the particle surface area increases with the amount of ALW increasing". Could the SMPS measurements provide quantitative evidence?

8. Page 13, line 4. Under high RH, the acidity, if any, is likely lower due to water dilution.

---

## Referee Comment (RC3) · Anonymous Referee #3 · 21 Apr 2019

General Comments:

This paper presents results from a series of simulation chamber experiments on the formation of secondary organic aerosol (SOA) from the photooxidation of furan. SOA yield was found to vary with both VOC/NOx ratio and relative humidity. Some information on the chemical composition of the SOA is presented, along with possible chemical mechanisms for the generation of several of the identified species. The experiments appear to have been conducted appropriately, however there are some aspects of the data interpretation that need to be clarified and/or improved prior to publication.

1. The effect of relative humidity in increasing SOA yield seems to be very similar to that reported for p-xylene by Healy et al. (2009), who proposed that the increase in relative humidity results in higher levels of HONO formation in the chamber which leads

to increased OH concentration, a faster p-xylene decay rate, and higher aerosol mass yields. Could the same effect be happening here? Was there a change in the rate of decay of furan as the relative humidity was increased through experiments 6-12?

2. A similar kinetic effect may also be occurring in experiments 1-5 where the VOC/NOx ratio was varied. Nitrous acid (HONO) is the main source of OH produced by heterogeneous reaction of nitrogen dioxide with water at the walls of the reactor. When NOx is increased, it is mainly in the form of NO2 and this may result in more OH formation, faster furan oxidation and more SOA formation. The authors should check the rate of furan decay in this subset of experiments and report/interpret accordingly.

Minor Comments:

1. The abstract is written in a generic manner and should be re-written to contain information specific to this work. For example, it is stated that the reaction conditions affected SOA yields. But it should state something like "varying VOC/NOx ratios over the range 48 to 8 cause SOA yields to increase from 0.04% to 0.5% under dry conditions". Some similar statements should be used to report the influence of relative humidity.

2. On page 2 (line 27), it is mentioned that several studies have previously investigated SOA formation from furan, but they appear to focus only on kinetic and mechanistic aspects.

3. On page 3 (line 26), it is mentioned that sea salt particles are the second most abundant particles in the atmosphere. This statement is seemingly used to justify the use of NaCl as seed particles, while it is more common to use ammonium sulfate as seeds in SOA formation experiments. Given that furan is a product of biomass burning and is also more likely to be released in urban environments than marine environments, the use of NaCl as seeds seems rather odd. The authors should provide some more reasons why NaCl particles were used as seeds.

References:

Robert M Healy, Brice Temime, Kristina Kuprovskyte, John C Wenger; Effect of Relative Humidity on Gas/Particle Partitioning and Aerosol Mass Yield in the Photooxidation of p-Xylene; Environ. Sci. Technol., 43, 1884-1889, 2009.

---

## Author Comment (AC1) · 6 Aug 2019

We thank the Referee for the insightful comments. We have revised our manuscript according to the suggestions of the Referee's comments and our responses to the comments are as follows. The Referee's comments are in black, authors' responses are in blue, and changes to the manuscript are in red color text.

This study investigates the formation of secondary organic aerosols from the photooxidation of furan at different NOx and RH levels. SOA yields were measured using NaCl seeds to provide surface area for the partitioning of SOA-forming vapors. The chemical composition, in particular organic functional groups and a selection of molecular products, was characterized by FTIR and ESI-MS. The authors found a strong dependence of the measured SOA concentration, mass yield, and the intensity of individual functionalities on the initial VOC/NOx ratios and RH levels in a series of experiments conducted, and suggested that NOx and RH play an important role in the SOA formation from furan oxidation by altering the chemical pathways, e.g., aqueous phase chemistry, that essentially lead to SOA. This conclusion, however, is heavily drawn from the inadequate data analysis and interpretation, and lacks fundamental understanding of the predominant chemistry that occurs in the chamber experiments performed.

NOx dependence of SOA yields

In this study, photolysis of NO2 was used to generate O3, which further undergoes photolysis and reaction with H2O to generate OH radicals. The initial VOC/NOx ratio in the performed experiments ranges from ~7 to ~48. With the presence of hundreds of ppb levels of furan at the beginning of the experiment, furan is not completely oxidized at the end and the measured SOA mass is mostly composed of the very first few generations of oxidation products. As the initial VOC/NOx ratio decreases, more NO2 will be available for the formation of O3 and consequently OH radicals. **The observed 'NOx-effect' here is essentially the OH effect: higher OH levels result in more furan consumed, thus producing more SOA mass and higher SOA yield.**

**This OH effect on SOA production has been well studied and understood in the community**.

**Author reply:**

In our experiments, the OH radical was indeed produced during the photooxidation of furan instead of being added before the experiments started. But the original intension of our study was not to determine the OH effect on furan SOA formation. As indicated previously, SOA yields increase with increasing NOx concentration at low NOx levels and then decrease at higher NOx concentration (Sarrafzadeh et al., 2016a; Loza et al., 2014; Hoyle et al., 2011; Chan et al., 2010). All these studies demonstrated that the increase of SOA yield at low NOx levels was attributed to an increase of OH concentration. After eliminating the effect of OH concentration on the SOA mass growth, the SOA yield only decreased with increasing NOx concentration.

To control the effect of OH concentration on furan SOA formation in the present work, four more experiments have been added with additional injection of $H_2O_2$ as OH precursor before the experiments started. The results suggested that there remains a positive correlation between SOA formation and NOx concentration as shown in Table S1 and Fig. 3. To further understand the effect of NOx level on SOA formation, four more experiments were carried out for HESI-Q Exactive-Orbitrap MS detection. The MS results revealed the formation of a number of cyclical hydroxyl nitrates and dihydroxyl dinitrates with low-volatility, which can significantly contribute to the SOA formation (Schwantes et al., 2019). To provide a better illustration of the NOx effect on SOA formation, the following changes have been made to the revised manuscript.

**Page 7, line 2:**

[revised manuscript text omitted]

RH dependence of SOA yields

Again, as H2O is used to generate OH radicals, higher RH levels result in more OH radicals, which lead to more SOA mass produced from furan photooxidation. **The**

**observed 'RH effect' is essentially another 'OH effect' by promoting the generation of OH radicals and accelerating the oxidation processes of furan**. The authors suggest that relative humidity affects the SOA yield through aqueous phase chemistry. However, the deliquescence point of sodium chloride is around 70% RH, below which the water content in the NaCl particles is close to zero, meaning that there would be minimal aqueous phase chemistry occurring in the particle phase.

**Author reply:**

The DRH and ERH of NaCl are about 75.5% and 47.6-46.3% (Gupta et al., 2015), respectively. NaCl seed aerosols were generated via atomization of NaCl aqueous solution with a constant-rate atomizer. The seed particles would be in the form of droplets after produced from the atomizer (Ge et al., 2016). Being dried through a self-made diffusion dryer, the NaCl seeds experienced efflorescence behavior during the dehydration process. It has been previously shown that SOA formation decreases both the ERH and DRH of the seed particles and results in the uptake of water by the particles (Liu et al., 2018; Takahama et al., 2007; Smith et al., 2012). There is a high possibility that the NaCl seeds will effloresce and deliquesce early after being coated by the new formed SOA. This assumption has been mentioned in the manuscript on Page 14, line 17. Additionally, we have monitored the aerosol liquid water content after the experiments. The details for the detection of ALW have been presented on Page 5, line 23 in our manuscript. Furthermore, as displayed in the manuscript, the ALW content has been indeed detected at 54% RH. Therefore, there is a high possibility that aqueous phase chemistry may play an important role in the SOA formation. This assumption was further confirmed by the results of HESI-Q Exactive-Orbitrap MS. The appearance of $m/z^+$ 185 and $m/z^-$ 262 demonstrated that aqueous phase reactions indeed took place under high RH conditions by aqueous phase reaction of the hydroxyfuranone (B) and 1,4-dialdehyde (D) (Strollo and Ziemann, 2013). Alternatively, the moist surface under high RH conditions is more favorable for the condensation of the products with carbonyl functional groups,

leading to the increasing production of SOA formation. As shown in Fig. 8, the intensities of multifunctional hydroxyl nitrates and dihydroxyl dinitrate (m/z⁻ 146, 225) exhibited positive correlations with RH conditions. Slight peak intensity increases of m/z+ 85 and 101 products were also observed under high RH conditions, indicating that the gas-particle phase partitioning of low-volatility compounds was enhanced at these conditions. The discussion concerning the effect of RH on furan SOA formation has been rewritten to provide more information.

**Page 9, line 28:**

"It is worth note that under high RH conditions, as shown in Fig. 1, the $NO_2$ hydrolysis (reaction (R11)) can generate nitrous acid (HONO), which has been considered as a major source of OH. As indicated in Fig. 4, the SOA yields obtained in the present work clearly show a gradual increase with RH. Also shown in Fig. S3 is the dependence of OH and furan concentrations on RH during the experiments determined from the decay of furan using a reaction rate coefficient of $k$(OH+furran) = 4.01× $10^{-11}$ $cm^3$ molecules$^{-1}$ s$^{-1}$ (Atkinson et al., 1983). It is therefore probable that the increase of RH results in high levels of HONO formation in the chamber, which leads to an increase in OH concentration, a faster furan decay rate, and higher aerosol mass yields. This result is in reasonable good agreement with previous studies, which proposed that the amount of products that can partition into the particle phase increases with the increasing rate of hydrocarbon oxidation (Healy et al., 2009; Chan et al., 2007). Moreover, the increasing RH might also enhance the SOA formation due to the fact that the functionalized gas phase components were more favoured to condense on the surface of wet particles (Liu et al., 2019b)."

**Page 13, line 3:**

[revised manuscript text omitted]

SOA measurements at high RH levels

It is well known that using DMA to measure aerosol size distribution and mass leadings under high RH is subject to many certainties, e.g., arcing at high voltage caused by high water content in the aerosols. While the authors used a diffusion drier in front of the DMA inlet, which could certainly minimize the arcing effect that interferes the measurement of big particles, the drying efficiency was not characterized. Have the authors measured the RH of the aerosol flow upon the exit of the diffusion drier? Did aerosols generated under high RH (e.g., 80%) still carry a certain amount of water after drying? Additionally, the authors need to consider how the drying processes affect the repartitioning of water-soluble components between gas and particle phases in order to obtain an accurate SOA yield.

**Author reply:**

Indeed, the Nafion dryer was added to determine the liquid water content in the aerosols under high RH after the reaction ended. After modifying to the dry mode, the humid air in SMPS was quickly replaced by dry air through venting the sheath air at 5 L min$^{-1}$, and then the dry aerosol was measured by SMPS. The ALW was determined by the difference of the particle mass concentrations before and after the modification of the dry mode. After the dry mode treatment, the RH in the sampling air and sheath air reduced to 10 % and 7 %, respectively. The drying process for ALW determination is based on a widely used method developed by Engelhart (Engelhart et al., 2011), which has been proven to remove 90% of the water vapor. Consequently, the aerosols treated after the drying process were thought to carry seldom water content. We agree with the Referee that there is a possibility of the water-soluble components repartitioning betweent gas and particle phases. The fact that the SOA concentration for high RH conditions were slightly underestimated due to the ALW measurement. Thus, we added some sentences to reply comment on how the ALW meaurement affects our results on Page 5 in the revised manuscript.

"It should be noted that the dissloved water-soluble species would evaporate back into

the gas phase during the ALW measurement when the aerosol water is removed. In fact, the repartitioning of water-soluble components between gas and particle phases was not taken into consideration. The SOA concentrations for high RH conditions were slightly underestimated, but the underestimation is extremely low and can be negligible."

Treatment of wall losses

The particle and vapor wall loss rates are chamber specific quantities that depend on a number of different parameters, i.e., the chamber size (volume to surface area ratio), the wall materials, the humidity in the enclosure air that affects the static charges on the chamber wall surface, and the mixing conditions (static or active mixing), among many others. The interaction patterns of particles with the chamber walls have been well studied for decades, and the particle wall loss rate has been found to vary substantially, by orders of magnitude, among different chamber environments. That the authors simply took the particle wall loss parameterizations obtained in other chambers to correct their own experiment would no doubt introduce significant uncertainties in their SOA mass measurements, resulting in unreliable SOA yield calculations.

**Author reply:**

We agree with the Referee that the wall loss rates of different chambers are different and the difference is dependent on many parameters, such as, the chamber size, the wall materials, and the chamber environments. The particle and vapor wall loss parameterizations used in this study are based on a previous study carried out in the same chamber reactor in our lab. Besides the chamber size and wall material, the experimental environments of these two studies are also similar: i) both studies were conducted in the ~5%-80% humidity range; ii) in both studies, NaCl seeds were added at the beginning of each experiment to provide sufficient seed surface area to limit the effects of vapor wall losses; iii) during all experiments in these two studies, two

ironing air blowers were around the reactor to get rid of the electric charge on the surface of the reactor. Consequently, the citation of wall loss parameterizations detected from the same chamber reactor and similar experimental experiments will not introduce much uncertainties in the present study. After careful analysis of our experiments, we believe that our results are reliable and credible. To clarify the statement, we modified the sentences in the revised manuscript. The new one reads as follows:

"The wall loss rate constants for $O_3$, NOx and aerosol particles were $3.3\times10^{-7}$ $s^{-1}$, $4.1\times10^{-7}$ $s^{-1}$, and $3.6\times10^{-5}$ $s^{-1}$, respectively, which were detected from our previous study conducted in the same set-up and similar experimental conditions (Ge et al., 2017a)."

[revised manuscript text omitted]

---

## Author Comment (AC2) · 6 Aug 2019

We greatly value the careful reading and the detailed comments provided by the Referee. The responses to the Referee's comments in our direct reply and within the revised manuscript are provided below. The original comments from Referee are in black, our replies are in blue and the tracked changes in the main manuscript are in red color text.

This manuscript aimed to study SOA formation from furan under photooxidation conditions with varied NOx and RH. SOA mass, O3 concentration, and SOA composition were carefully measured. The authors concluded that furan photooxidation is dominated by RO2 + NO chemistry that leads to formation of carbonyl-rich products. SOA formation was found to enhance under higher relative humidity and higher VOC/NOx concentration. The results are clearly presented, but there are a few important issues that need to be addressed before becoming publishable. My major comment is that the manuscript did not provide sufficient chemistry insights given the suite of instruments used.

**Author reply:**

We greatly appreciate the thoughtful and helpful comments proposed by the Referee. To have a more sufficient chemistry insights into the furan SOA formation, we have added five more experiments under a series of NOx and RH conditions for the analysis of an improved chemical identification instrument. According to the latest results that sufficiently eliminate the interference of artifacts on the Exactive-Orbitrap MS, few parts of the discussion have been modified. Spectra results and reaction mechanism have been updated as well. Further discussion is given in later sections.

**Major:**

1. The authors claims that the O3 maximum concentration decreases with increasing RH likely due to that high RH favors the partitioning of the NOx reservoir, RNO2 and RONO2 into the particle phase. Thus the release of NOx is limited. However, it is

unclear how much more pronounced is the RNO2 and RONO2 uptake at higher RH and whether this enhancement is sufficient to affect O3 formation. These species need to be very soluble to show distinct partitioning behavior under different RH. What are their possible structures and could the Henry's law constants be estimated? Were any RNO2 or RONO2 species enhanced in the SOA samples under higher RH? From the ESI-MS results, the authors said that the SOA composition is similar between higher RH and low RH. Does this contradict what the authors concluded earlier? You have a max SOA yield of 5% and only a small fraction of that is RNO2 or RONO2. Thus are they important enough to affect gas-phase NOx concentration. Evidence is needed. Alternatively, could wall loss of the RNO2 and RONO2 under high RH better explain the observation?

**Author reply:**

According to the updated HESI-Q Exactive-Orbitrap MS results in the revised manuscript, the intensities of multifunctional hydroxyl nitrates and dihydroxyl dinitrate (m/z- 146, 225) were indeed enhanced under high RH conditions (Fig. 8). As demonstrated by a previous study, these multifunctional hydroxyl nitrates and dihydroxyl dinitrate have low volatilities and can partition into the particle phase (Schwantes et al., 2019). It should also be noted that some products generated from the furan photooxidation have C=C bonds conjugated with the carbonyl and acid groups. These products can absorb strongly in the ~200-300 nm range and contribute to the signal of O3 concentration detected by O3 analyzer (Strollo and Ziemann, 2013). However, under high RH conditions, these carbonyl-rich products were favourable to condense on the moist surface of particles and thus lowered the O3 concentration detected by  $O_3$  analyzer. This assumption is reasonable since the intensities of m/z+ 85 and 101 carbonyl-rich products were observed to increase when RH increased from dry condition to 30%. However, the intensities of the assigned carbonyl-rich products exhibited a slight decrease under higher RH conditions. This decreasing trend resulted from the aqueous phase reactions of hydroxyfuranone  $(m/z^+ 101)$  with 4-dialdehyde

 $(m/z^+ 85)$ , leading to the appearance of the generated hemiacetal compound  $(m/z^+ 185)$ . In addition, there is a possibility that the wall losses of the organonitrates species would inhibit the O3 formation under high RH as suggested by the Referee. To provide more information about the decreased O3 concentration under high RH, the following changes have been made to the revised manuscript.

**Page 7, line 28:**

"However, it should be noted that the apparent high  $O_3$  concentration monitored by the  $O_3$  analyzer was due to species other than  $O_3$  and that have absorption around 254 nm. The generated gas-phase reaction products that have carbonyl, carboxylic acid, and ester groups conjugated with C=C bonds can absorb strongly in the ~200-300 nm range and, accordingly, contribute to the apparent intense signal of  $O_3$  detected by the  $O_3$  monitor (Strollo and Ziemann, 2013)."

**Page 8, line 16:**

"Additionally, the slight change in O3 maximum concentration under different RH conditions may also be caused by the consumption of gas-phase reaction products that contain functional groups conjugated with C=C bonds and respond to the O3 analyzer. This assumption is reasonable because these carbonyl-rich products were favourable to condense on the moist surface of particles and thus lowered the O3 concentration detected by the O3 analyzer, which has been further confirmed by the MS results with enhanced intensities of corresponding organonitrates. The wall loss of the organonitrates species would also explain the decreasing trend of O3 concentration as RH increases."

Figure 8: Selected background-subtraction HESI-Q Exactive-Orbitrap MS results of SOA in both negative (blue) and positive (pink) ion modes from the photooxidation of furan under different RH conditions.

Table 3. Ion peaks with the assigned compounds observed in the HESI-Q Exactive-Orbitrap MS. Proposed assignments are based on the formula from HESI-Q Exactive-Orbitrap MS.

| Ion mode             | No | Mass     | Ion Mode           | Ion                                                         | Delta  | Proposed Structure                                                                                                                                                                                                                                                                                                                                                                                                                                                                                                                                                                                                                                                                                                                                                                                                                                                                                                                                                                                                                                                                                                                                                                                                                                                                                                                                                                                                                                                                                                                                                                                                                                                                                                                                                                                                                                                                                                                                                                                                                                                                                                                                                                                                                                                                                                                                                                                                                                                                                                                                                                                                                                                                                                                                                                                                                                                                                                                                                                                                                                                                                                                                                                                                                                                                                                                                                                                                                              |  |
|----------------------|----|----------|--------------------|-------------------------------------------------------------|--------|-------------------------------------------------------------------------------------------------------------------------------------------------------------------------------------------------------------------------------------------------------------------------------------------------------------------------------------------------------------------------------------------------------------------------------------------------------------------------------------------------------------------------------------------------------------------------------------------------------------------------------------------------------------------------------------------------------------------------------------------------------------------------------------------------------------------------------------------------------------------------------------------------------------------------------------------------------------------------------------------------------------------------------------------------------------------------------------------------------------------------------------------------------------------------------------------------------------------------------------------------------------------------------------------------------------------------------------------------------------------------------------------------------------------------------------------------------------------------------------------------------------------------------------------------------------------------------------------------------------------------------------------------------------------------------------------------------------------------------------------------------------------------------------------------------------------------------------------------------------------------------------------------------------------------------------------------------------------------------------------------------------------------------------------------------------------------------------------------------------------------------------------------------------------------------------------------------------------------------------------------------------------------------------------------------------------------------------------------------------------------------------------------------------------------------------------------------------------------------------------------------------------------------------------------------------------------------------------------------------------------------------------------------------------------------------------------------------------------------------------------------------------------------------------------------------------------------------------------------------------------------------------------------------------------------------------------------------------------------------------------------------------------------------------------------------------------------------------------------------------------------------------------------------------------------------------------------------------------------------------------------------------------------------------------------------------------------------------------------------------------------------------------------------------------------------------------|--|
|                      |    | (m/z)    |                    | Formula                                                     | (amu)  | Troposed Substance                                                                                                                                                                                                                                                                                                                                                                                                                                                                                                                                                                                                                                                                                                                                                                                                                                                                                                                                                                                                                                                                                                                                                                                                                                                                                                                                                                                                                                                                                                                                                                                                                                                                                                                                                                                                                                                                                                                                                                                                                                                                                                                                                                                                                                                                                                                                                                                                                                                                                                                                                                                                                                                                                                                                                                                                                                                                                                                                                                                                                                                                                                                                                                                                                                                                                                                                                                                                                              |  |
| Positive
ion mode | 1  | 85.0018  | [M+H] + | $C_4H_5O_2{}^+$                                             | -0.027 | 0                                                                                                                                                                                                                                                                                                                                                                                                                                                                                                                                                                                                                                                                                                                                                                                                                                                                                                                                                                                                                                                                                                                                                                                                                                                                                                                                                                                                                                                                                                                                                                                                                                                                                                                                                                                                                                                                                                                                                                                                                                                                                                                                                                                                                                                                                                                                                                                                                                                                                                                                                                                                                                                                                                                                                                                                                                                                                                                                                                                                                                                                                                                                                                                                                                                                                                                                                                                                                                               |  |
|                      | 2  | 101.0894 | $[M+H]^+$          | $C_4H_5O_3^+$                                               | 0.066  |                                                                                                                                                                                                                                                                                                                                                                                                                                                                                                                                                                                                                                                                                                                                                                                                                                                                                                                                                                                                                                                                                                                                                                                                                                                                                                                                                                                                                                                                                                                                                                                                                                                                                                                                                                                                                                                                                                                                                                                                                                                                                                                                                                                                                                                                                                                                                                                                                                                                                                                                                                                                                                                                                                                                                                                                                                                                                                                                                                                                                                                                                                                                                                                                                                                                                                                                                                                                                                                 |  |
|                      | 3  | 185.0504 | $[M+H]^+$          | $\mathrm{C_8H_9O_5^+}$                                      | 0.006  |                                                                                                                                                                                                                                                                                                                                                                                                                                                                                                                                                                                                                                                                                                                                                                                                                                                                                                                                                                                                                                                                                                                                                                                                                                                                                                                                                                                                                                                                                                                                                                                                                                                                                                                                                                                                                                                                                                                                                                                                                                                                                                                                                                                                                                                                                                                                                                                                                                                                                                                                                                                                                                                                                                                                                                                                                                                                                                                                                                                                                                                                                                                                                                                                                                                                                                                                                                                                                                                 |  |
| Negative
ion mode | 4  | 146.0161 | [M-H] - | C4H4O5N-                                                    | 0.007  | $HO \qquad O_2NO \ O_2NO \ O_2NO \$ |  |
|                      | 5  | 225.0125 | [M-H] - | $C_4H_5O_9N_2^-$                                            | 0.012  |                                                                                                                                                                                                                                                                                                                                                                                                                                                                                                                                                                                                                                                                                                                                                                                                                                                                                                                                                                                                                                                                                                                                                                                                                                                                                                                                                                                                                                                                                                                                                                                                                                                                                                                                                                                                                                                                                                                                                                                                                                                                                                                                                                                                                                                                                                                                                                                                                                                                                                                                                                                                                                                                                                                                                                                                                                                                                                                                                                                                                                                                                                                                                                                                                                                                                                                                                                                                                                                 |  |
|                      | 6  | 262.0132 | [M-H] - | C 8 H 8 O 9 N - | -0.007 |                                                                                                                                                                                                                                                                                                                                                                                                                                                                                                                                                                                                                                                                                                                                                                                                                                                                                                                                                                                                                                                                                                                                                                                                                                                                                                                                                                                                                                                                                                                                                                                                                                                                                                                                                                                                                                                                                                                                                                                                                                                                                                                                                                                                                                                                                                                                                                                                                                                                                                                                                                                                                                                                                                                                                                                                                                                                                                                                                                                                                                                                                                                                                                                                                                                                                                                                                                                                                                                 |  |

2. The ESI-MS results presented grouped ions and observed m/z 200-299 as the most abundant group. What does this mean? What are the major chemical formulas observed in this range? I think a lot more discussion of the ESI-MS results could be included here. For example:

(1) Could the authors group all nitrogen-containing species and discuss their presence?

(2) What are the ranges of O:C ratio and oxidation state?

(3) Any variation as the furan/NOx and RH changed?

(4) For soluble species uptaking onto aqueous particles, they will likely oligomerize.Do the authors see oligomers in the ESI-MS?

(5) There is a substantial decrease of large molecules under lower furan/NOx ratio and higher RH in the negative ion mode (but not in the positive ion mode). What does that suggest?

(6) what is the ion in the negative ion mode around m/z 380?

In the current form, there is very little discussion (page 11, line 16-24, page 12, line 2-7, and page 13, line 8-11)

**Author reply:**

A simple discussion of the m/z 200-299 parts may be confusing for a better understanding of the SOA components. In our revised manuscript, this part has been deleted and a detailed introduction of the updated Exactive-Orbitrap MS results was presented. Specifically, the nitrogen-containing species have been grouped and discussed in the revised manuscript as suggested by the Referee. Generally, three kinds of nitrogen-containing organic species were observed by the HRMS with  $m/z^-$  of 146.0161, 225.0125 and 263.0132 being detected in negative ion mode. The assignments of these ion peaks, the molecular weights, and proposed structures are summarized in Table 3. However, owing to the limits of our techniques, the ranges of O:C and oxidation state cannot be provided here. For carbonyl-rich species, these products were favourable to condense on the moist surface of particles and took part in further reactions. According to the mass results, the observed hemiacetal compound  $(m/z^+ 185)$  was generated from the oligomerizations of hydroxyfuranone  $(m/z^+ 101)$ and 4-dialdehyde  $(m/z^+ 85)$ . After removing the interference both from the sampling filter and residues from the detection instruments, new experiments have been conducted. The collected particles have been re-detected and the detected products were re-assigned in the revised manuscript. Moreover, the ESI-MS results have been significantly expanded as following:

**Page 11, line 21:**

[revised manuscript text omitted]

---

## Author Comment (AC3) · 6 Aug 2019

We thank the Referee for the very insightful comments. We have revised our manuscript according to the suggestions of the Referee's comments and our responses to the comments are as follows. Referee's comments are in black, authors' responses are in blue, and changes to the manuscript are in red color text.

This paper presents results from a series of simulation chamber experiments on the formation of secondary organic aerosol (SOA) from the photooxidation of furan. SOA yield was found to vary with both VOC/NOx ratio and relative humidity. Some information on the chemical composition of the SOA is presented, along with possible chemical mechanisms for the generation of several of the identified species. The experiments appear to have been conducted appropriately, however there are some aspects of the data interpretation that need to be clarified and/or improved prior to publication.

1. The effect of relative humidity in increasing SOA yield seems to be very similar to that reported for p-xylene by Healy et al. (2009), who proposed that the increase in relative humidity results in higher levels of HONO formation in the chamber which leads to increased OH concentration, a faster p-xylene decay rate, and higher aerosol mass yields. Could the same effect be happening here? Was there a change in the rate of decay of furan as the relative humidity was increased through experiments 6-12?

**Author reply:**

According to the Referee's suggestions, we have added a figure of relationships between relative humidity and OH concentration as well as the degradation profile of furan under different RH conditions. The effects of RH conditions on OH concentration and SOA formation found in the present study are similar to those reported by Healy et al. An increasing trend of OH concentration and a faster decay rate of furan were observed as RH increased, as shown in Fig. S3. It is possible that the faster rate of gas phase oxidation under higher OH concentrations would lead to

the generation of less volatile compounds as presented previously (Chan et al., 2007). This part has been revised on page 9 and page 13.

**Page 9, line 28:**

"It is worth note that under high RH conditions, as shown in Fig. 1, the $NO_2$ hydrolysis (reaction (R11)) can generate nitrous acid (HONO), which has been considered as a major source of OH. As indicated in Fig. 4, the SOA yields obtained in the present work clearly show a gradual increase with RH. Also shown in Fig. S3 is the dependence of OH and furan concentrations on RH during the experiments determined from the decay of furan using a reaction rate coefficient of $k$(OH+furan) = $4.01 \times 10^{-11}$ $cm^3$ molecules$^{-1}$ s$^{-1}$ (Atkinson et al., 1983). It is therefore probable that the increase of RH results in high levels of HONO formation in the chamber, which leads to an increase in OH concentration, a faster furan decay rate, and higher aerosol mass yields. This result is in reasonable good agreement with previous studies, which proposed that the amount of products that can partition into the particle phase increases with the increasing rate of hydrocarbon oxidation (Healy et al., 2009; Chan et al., 2007). Moreover, the increasing RH might also enhance the SOA formation due to the fact that the functionalized gas phase components were more favoured to condense on the surface of wet particles (Liu et al., 2019). "

**Page 14, line 8:**

"As shown in Fig. S3, an increase in RH leads to higher OH concentrations resulting from higher HONO levels generated by the reaction of $NO_2$ with $H_2O$. Previously, Anglada et al. confirmed, using quantum mechanical calculations, that the water component could increase the OH production (Anglada et al., 2011). The positive correlation between initial water vapour concentration and OH concentration has also been previously observed experimentally (Healy et al., 2009; Tillmann et al., 2010). Additionally, Healy et al. have also reported that increasing OH concentration promoted the decay of VOC and enhanced SOA formation (Healy et al., 2009). Similarly, in the present work, a faster decay rate of furan was also observed as RH

increased, as shown in Fig. S3. It is possible that the faster rate of gas phase oxidation under higher OH concentrations will lead to the generation of less volatile compounds as presented previously (Chan et al., 2007). A higher OH concentration promotes oxidation reactions, influences the distribution of organic products, and facilitates the SOA formation (Sarrafzadeh et al., 2016)."

[Figure]

Figure S3: Relationship between relative NOx concentrations, humidity, and steady OH concentration in the furan photooxidation experiments performed under different NOx levels and RH conditions. The OH concentration was calculated based on $[OH]=$ $\ln \frac{[furan]_0}{[furan]_t} /kt$, $k$=4.01× $10^{-11}$ cm$^3$ molecules$^{-1}$ s$^{-1}$ (Atkinson et al., 1983).

2. A similar kinetic effect may also be occurring in experiments 1-5 where the VOC/NOx ratio was varied. Nitrous acid (HONO) is the main source of OH produced by heterogeneous reaction of nitrogen dioxide with water at the walls of the reactor. When NOx is increased, it is mainly in the form of NO2 and this may result in more OH formation, faster furan oxidation and more SOA formation. The authors should check the rate of furan decay in this subset of experiments and report/interpret accordingly.

**Author reply:**

As suggested by the Referee, the relationship between NOx level and OH concentration as well as the degradation profile of furan have been added in Fig. S3. Experiments conducted under different NOx levels indicate that the OH concentration was controlled by the NOx level if there were no additional OH precursors added before the experiment. As shown in Fig. S3, the OH concentration exhibits a gradual increase with the increase of NOx concentration and there appears to be a correlation between NOx concentration, OH concentration and SOA yield. Therefore, the increase of SOA yield was attributed to an increase of OH concentration, which was affected by OH recycling (reaction (R6), Fig. 1) and thus contributed to the enhancement of SOA formation.

**Page 9, line 8:**

[revised manuscript text omitted]

2. On page 2 (line 27), it is mentioned that several studies have previously investigated SOA formation from furan, but they appear to focus only on kinetic and mechanistic aspects.

**Author reply:**

We have revised this sentence as follows:

"Although the determination of kinetics and products of furan oxidation has been performed (Cabanas et al., 2004; Liljegren and Stevens, 2013; Tapia et al., 2011), the influence of several factors including NOx level and relative humidity on SOA formation from furan has not been well examined."

3. On page 3 (line 26), it is mentioned that sea salt particles are the second most abundant particles in the atmosphere. This statement is seemingly used to justify the use of NaCl as seed particles, while it is more common to use ammonium sulfate as seeds in SOA formation experiments. Given that furan is a product of biomass burning and is also more likely to be released in urban environments than marine

environments, the use of NaCl as seeds seems rather odd. The authors should provide some more reasons why NaCl particles were used as seeds.

**Author reply:**

We agree that ammonium sulfate is more commonly used as seeds in smog chamber simulations compared to NaCl. But concerning the strong absorption of ammonium sulfate in mid-infrared region, the addition of ammonium sulfate would interfere the assignment of the functional groups in the FTIR spectra. Specifically, the following figure shows the characteristic peaks of ammonium sulfate: the stretching modes of the $SO_4^{2-}$ ion are at 1089 and 975 $cm^{-1}$; the broad band between 3400 and 2800 $cm^{-1}$ is assigned to $NH_4^+$ stretching. However, NaCl does not have the absorption band in mid-infrared region. Consequently, we decided to choose NaCl as seeds in the present experiment. To remove any confusion regarding the use of NaCl seed particles, the statement has been modified as:

[Figure]

"All the experiments were conducted in the presence of NaCl seed particles, which acted as the nuclei and provided sufficient seed surface area at the beginning of the reaction to suppress the effects of vapour wall losses of semi/low-volatility species."